# Wafer-scale integration of sacrificial nanofluidic chips for detecting and manipulating single DNA molecules

Chao Wang[1,2,*], Sung-Wook Nam[1,*,†], John M. Cotte[1], Christopher V. Jahnes[1], Evan G. Colgan[1], Robert L. Bruce[1], Markus Brink[1], Michael F. Lofaro[1], Jyotica V. Patel[1], Lynne M. Gignac[1], Eric A. Joseph[1], Satyavolu Papa Rao[1,†], Gustavo Stolovitzky[1,3], Stanislav Polonsky[1,†] & Qinghuang Lin[1]

Wafer-scale fabrication of complex nanofluidic systems with integrated electronics is essential to realizing ubiquitous, compact, reliable, high-sensitivity and low-cost biomolecular sensors. Here we report a scalable fabrication strategy capable of producing nanofluidic chips with complex designs and down to single-digit nanometre dimensions over 200 mm wafer scale. Compatible with semiconductor industry standard complementary metal-oxide semiconductor logic circuit fabrication processes, this strategy extracts a patterned sacrificial silicon layer through hundreds of millions of nanoscale vent holes on each chip by gas-phase Xenon difluoride etching. Using single-molecule fluorescence imaging, we demonstrate these sacrificial nanofluidic chips can function to controllably and completely stretch lambda DNA in a two-dimensional nanofluidic network comprising channels and pillars. The flexible nanofluidic structure design, wafer-scale fabrication, single-digit nanometre channels, reliable fluidic sealing and low thermal budget make our strategy a potentially universal approach to integrating functional planar nanofluidic systems with logic circuits for lab-on-a-chip applications.

[1] IBM T.J. Watson Research Center, 1101 Kitchawan Road, PO Box 218, Yorktown Heights, New York 10598, USA. [2] School of Electrical, Computer and Energy Engineering, and Biodesign Center for Molecular Design & Biomimetics, Arizona State University, Tempe, Arizona 85287, USA. [3] Department of Genetics and Genomics Sciences, Icahn School of Medicine at Mount Sinai, New York, New York 10029, USA. * These authors contributed equally to this work. † Present address(es): Department of Molecular Medicine, School of Medicine, Kyungpook National University, 807 Hoguro, Bukgu, Daegu 41404, South Korea (S.-W.N.); 257 Fuller Road, SUNY Poly SEMATECH, Albany, New York 12203, USA (S.P.R.); Samsung R&D Institute Russia, 12/1 Dvintsev street, Moscow 127018, Russia (S.P.). Correspondence and requests for materials should be addressed to C.W. (email: wangch@asu.edu) or to G.S. (email: gustavo@us.ibm.com) or to S.P. (email: s.polonsky@samsung.com) or to Q.L. (email: qhlin@us.ibm.com).

Advanced nanofluidic systems[1,2], for example, nanochannels[3,4] and nanopores[5,6], have enabled manipulation of deoxyribonucleic acid (DNA) biopolymers with unprecedented control by exploiting the complex fluidic dynamic interactions with nanostructures, such as nanoconfinement induced stretching[3,4], collision induced straddling and stretching[7], and lateral displacement induced separation[8]. These systems have demonstrated effective sorting[4,8–10], sensing[5,6,11–14] and analysis[15,16] at low sample concentration and even single-molecule level. In addition, on-chip integration with electronic components[11,17] can significantly improve the nanofluidic functionality and reduce the device footprint, which are essential to realizing ubiquitous, compact, high-sensitivity and cost-effective biomolecular sensors. However, unlike complementary metal-oxide semiconductor (CMOS) chips which utilize a relatively small range of feature dimensions at one particular level, nanofluidic chips must integrate more sophisticated three-dimensional architectures incorporating vacant and sealed nanostructures with dimensions spanning several orders of magnitude to optimally manipulate and detect biomolecules. The stringent requirement of reliably forming and sealing complex and small nanostructures makes it very challenging to fabricate nanofluidic chips over a wafer scale by current CMOS processes, and thus has seriously hindered electronics integration.

Conventionally, nanofluidic chip fabrication processes generally exploit one or a combination of the following approaches (Supplementary Note 1 and Supplementary Fig. 1): selective sealing[18,19], wafer bonding[3,7,11,20] and sacrificial materials. Selective sealing methods demand special materials and nanostructure geometries (for example, height, width, shape and so on) to seal the nanofluidic channels without causing clogging. Wafer bonding to soft materials[11,20] generally suffer from leakage, low bonding strength, clogging due to polymer deformation and incompatibility with various chemicals; in comparison, bonding to rigid materials[3,7] requires a high-temperature annealing to achieve a strong bonding strength, and poses processing yield and metal integration challenges. Sacrificial approaches utilize a material 'to be sacrificed' patterned into a reverse image of the desired nanofluidic structures, and selectively extract this sacrificial material at a later stage of processing to form the nanofluidic system. However, thermal decomposition based extraction method[21] has serious risks of structural damage at elevated temperatures, and wet etching based extraction processes[22–24] are ineffective at nanometre scales and potentially destructive, because removing etched byproduct becomes exceedingly difficult and undesirable long processing time is needed[24].

So far, existing nanofabrication technologies face serious challenges in patterning single-digit nanometre features, producing complex planar fluidic structures, and also integrating metallic elements over a wafer scale. There is a strong demand to develop CMOS-compatible fabrication methods to bridge the gap between conventional CMOS-based electronic signal-processing platforms and biomolecular sensing systems.

Here we report a scalable fabrication strategy based on gas-phase etching of patterned sacrificial silicon (Si) layer using Xenon difluoride ($XeF_2$)[25,26] gas. $XeF_2$ etching of Si is a well-established technique with demonstrated compatibility with Si processing[25,27]. However, to date, its applications in complexed designed functional nanofluidics have been rather limited[28]. Our integration strategy was first disseminated in a conference abstract[29], but this paper provides for the first time detailed and complete discussions on the integration strategy, key challenging issues, fabrication results, single-digit nanometre channels, and single-molecule DNA straddling.

Our scalable and extensible integration strategy significantly differs from others $XeF_2$ based integration methods. First, previous demonstrations usually had large lateral dimensions, for example, from 10 to 100 μm (ref. 28). In contrast, the critical dimensions of our devices are about 3 orders of magnitude smaller. Second, single-molecule imaging and manipulation (for example, DNA stretching), which is important to biomolecular sensing, sorting and so on, has not been demonstrated previously in sacrificial nanochannels. Here we visualize fluorescently labelled single DNA molecule flow and verify their controlled stretching in nanochannels. Third, previous $XeF_2$ Si etching was only applied to simple geometries such as straight and long channels but not complex and functional fluidic network. In this work, we prototype two-dimensional fluidic network for controlled DNA fluidic dynamics and stretching. Fourth, conventional methods diffuse $XeF_2$ only through the fluidic ports to extract Si, which is inherently a time-consuming diffusion-limited process and strongly dependent on the channel dimensions. Differently, we rationally integrate venting holes to initiate the Si exaction in a massively parallel fashion, hence significantly reducing the process time, increasing the throughput and enabling complex fluidic design over large areas. Finally, we integrate our process completely on a 200 mm Si wafer processing platform, making it easier for our integration strategy to translate to high-volume production. The advantages of our approach include low thermal budget (room temperature), simple processing (gas phase process, free of tedious wetting and drying), minimal contamination (no solid or liquid chemical residues), fast etching rate (tens of micrometres per minute, 2 orders of magnitude faster than wet etching process[24]), extraordinarily high etching selectivity[25] (5 nm-thick $SiO_2$ can sustain etching of 300 μm silicon[30]), and compatibility with metallic elements and even logic circuit integration[26]. Therefore, our wafer-scale design and fabrication strategy is appealing to nanofluidic systems requiring both reliable control of structures at nanometre scale dimensions and high-volume, cost-effective wafer-scale manufacturing.

## Results

**Integration strategy for nanofluidics.** The critical fabrication steps are illustrated in Fig. 1, comprising: preparing a flat substrate with inlaid micrometre-thick sacrificial amorphous silicon (α-Si) microstructures (Fig. 1a,b), nanopatterning a thin (<100 nm) α-Si layer and connecting it to inlaid α-Si microstructures (Fig. 1c), and extraction of Si by $XeF_2$ etching through nanometre-scale vent holes and sealing the nanofluidic channels (Fig. 1d–f). Here α-Si was deposited at low temperatures (<150 °C) and was optimized for low film stress by varying the argon (Ar) gas pressure during deposition.

In our fabrication strategy, we incorporated several important features to build functional nanofluidic chips for manipulating DNA molecules[7]. First, the nanofluidic chips are compatible with high-resolution fluorescence imaging[7], by employing relatively long microfluidic structures (30 mm) that leave ample space for a microscope objective. Second, the fluidic structural dimensions on each chip span three orders of magnitude, from >10 μm for microfluidics to <10 nm for nanofluidics, in order to integrate different functionalities, that is, microstructures for fast fluidic transport and single-digit nanometre features for DNA manipulation and detection. Here we patterned sacrificial Si layers in two separate stages to fulfil these distinct requirements, and defined the critical structural dimensions by a three-level mixed lithography method, combining mid-ultraviolet contact lithography (MUV), deep-ultraviolet projection optical lithography (DUV), and electron beam

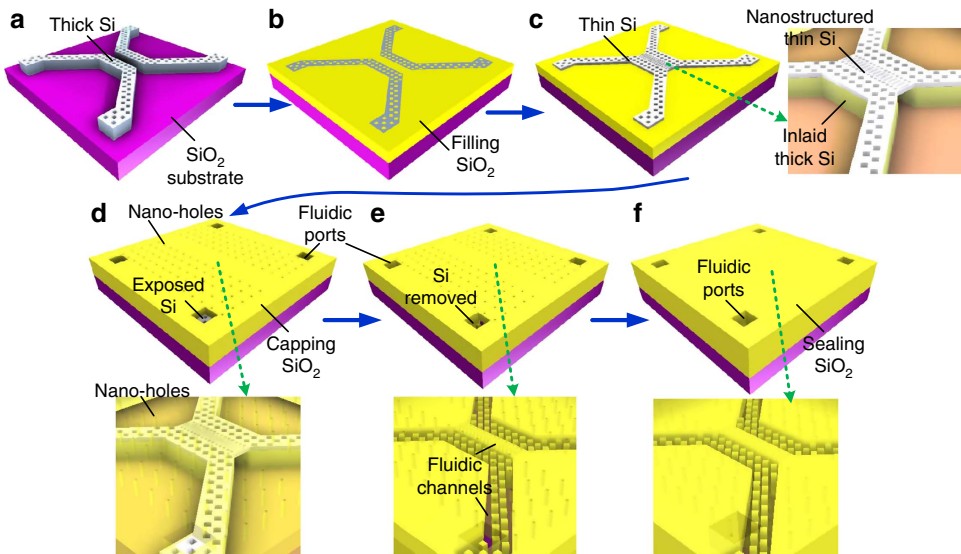

**Figure 1 | Fabrication scheme of Si sacrificial nanofluidic devices. (a)** Micrometre-thick Si microstructures fabricated on SiO$_2$/Si substrates.
**(b)** Planarized Si microstructures inlaid in a SiO$_2$ film (planarization SiO$_2$) after SiO$_2$ deposition and wafer polishing. **(c)** Thin Si nanostructures fabricated on top of inlaid Si microstructures. **(d)** Capping of Si fluidic structures by SiO$_2$. **(e)** Fluidic ports and venting nanoholes fabricated in capping SiO$_2$.
**(f)** Sealing venting nanoholes by depositing SiO$_2$ while keeping fluidic ports open. Critical three-dimensional structures (embedded α-Si features, venting holes and vacant channels) in figures **(c–f)** are better illustrated with the top films layers intentionally set as semi-transparent (as indicated by arrows).

lithography (EBL). Third, the micro- and nano-fluidic channels are patterned as two-dimensional meshes to ensure structural robustness with high-density support pillars, to minimize undesired dishing effects during surface planarization, and to facilitate DNA stretching[7].

**Inlaid microfluidic structure patterning**. First we prepared the 200-mm Si wafer with plasma enhanced chemical vapour deposition (PECVD) of silicon dioxide (SiO$_2$) and then deposited a blanket layer of sacrificial α-Si of nominally 2-μm-thick. The α-Si microstructures were patterned by MUV lithography and plasma etch (details in 'Methods' section). The α-Si microstructure must be planarized for the subsequent high-fidelity lithographic patterning. This was achieved by conformally coating α-Si with tetraethyl-orthosilicate (TEOS)-based 100 nm-thick SiO$_2$, followed by relatively less conformal yet more efficiently deposited 2-μm-thick PECVD SiO$_2$. After that, the topology of SiO$_2$ surface projected from the bottom α-Si patterns was removed by chemical mechanical polishing (CMP) to create the embedded α-Si structure with a flat surface (details in Supplementary Fig. 2, Supplementary Note 2 and 'Methods' section). After CMP, the surface height difference between interlaid Si mesh (10 μm wide) and SiO$_2$ pillar (20 μm wide) regions was found to be significantly reduced, from ∼2.2 μm (Supplementary Fig. 3) to ∼40 nm (Supplementary Fig. 4), which meets the requirements for the subsequent high-fidelity, reliable nanopatterning.

**Si nanostructure fabrication**. After creating the inlaid Si microstructures, we patterned thin α-Si structures (40 or 100 nm in our experiments) on top of the thick α-Si microfluidic structures (Fig. 2a–d), using a three-level mixed lithography (detailed scheme given in Supplementary Fig. 5 and Supplementary Note 3), that is, Nanometre scale (<100 nm) fluidic channels (nC) defined by EBL, submicron and nanometre scale functional nanochannels and nanopillars by DUV lithography for controlled hydrodynamic

DNA interactions and stretching at pillar interface[7], and micrometre scale fluidic channels (μC) defined by MUV lithography for fluidic connection and sample delivery. The three lithographic levels were patterned separately and then transferred together into the same hard mask (HM) layer (here a 20 nm PECVD SiO$_2$ layer) on top of the thin α-Si device layer.

In EBL, established CMOS fabrication infrastructure at IBM T.J. Watson research center was used to achieve high resolution and repeatable nanopatterning[31]. To achieve a good control of feature dimensions and maximize the process yields, monitor wafers were added to each batch and processed together with device wafers. This approach enabled us to optimize the recipes for the nanostructure patterning and minimize batch-to-batch and wafer-to-wafer variations. Selected scanning electron microscopy (SEM) images of the EBL-patterned nanofeatures are shown in Supplementary Fig. 6. Particularly, we used hydrosilsesquioxane (HSQ, 20 nm thick) resist on top of an organic planarization layer (OPL, 65 nm thick) for patterning critical nanostructures. Then a 450 nm-thick DUV resist was printed to form submicron nanopillars and nanochannels that are aligned and connected to the HSQ nanopatterns (Fig. 2a). The DUV resist and HSQ resist together provided the mask for etching the nanofluidic structures into the OPL underneath and subsequently into the SiO$_2$ HM layer (Fig. 2b). The α-Si layer was also micro-patterned by MUV lithography (Fig. 2c), therefore it was connected to thick inlaid α-Si vertically and the EBL/DUV printed α-Si nanostructures laterally to complete the fluidic paths from one port to another, protected the inlaid α-Si microstructures from plasma etch damage with slightly larger lateral dimensions (2.5 μm larger in our design), and turned the α-Si film into micro-meshes separated by SiO$_2$ posts to improve the structural integrity of the fluidics-sealing SiO$_2$ layer.

To minimize alignment errors in the three-level mixed lithography, all the three lithographic levels as well as the inlaid Si microstructures were aligned back to the same set of alignment marks, which were printed on the wafer before patterning inlaid Si microstructures. This alignment strategy was

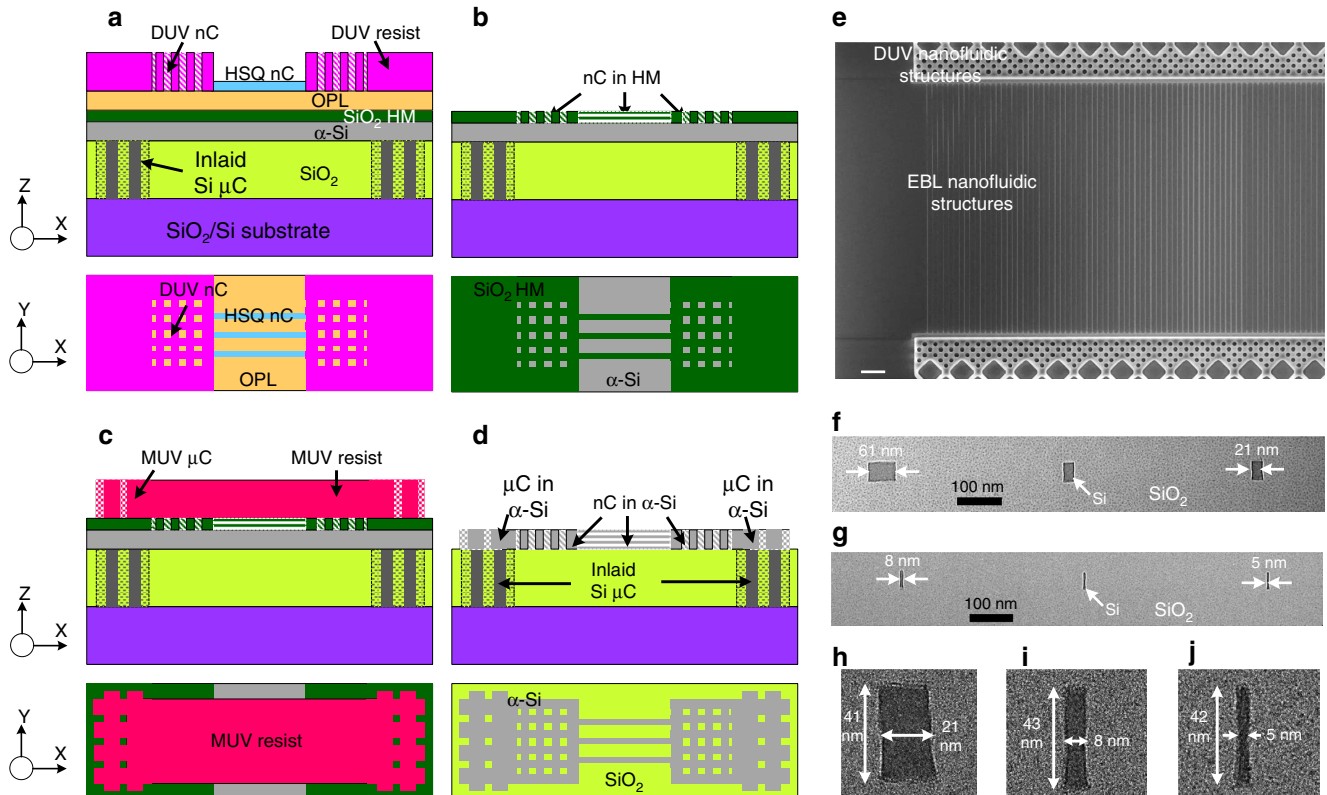

**Figure 2 | Three-level mixed lithography to fabricate Si sacrificial nanostructures.** (**a–d**) Fabrication scheme of patterning process (top: cross-sectional view in X–Z plane; bottom: top view in X–Y plane): (**a**) Combined nanopatterning of nanofluidic channel structures (nC) in deep ultraviolet (DUV) resist and EBL-defined hydrosilsesquioxane (HSQ) resist on an organic planarization layer (OPL)/SiO$_2$ HM (hard mask)/α-Si film stack, with inlaid Si microchannels (μC) embedded and planarized in the substrate; (**b**) Nanofluidic structures transferred to HM layer; (**c**) MUV-patterned μC in resist; (**d**) α-Si fluidic structures with critical dimensions defined by MUV, DUV and EBL. (**e-j**) Fabricated nanofluidic structures: (**e**) Scanning electron microscope (SEM) image showing well-aligned DUV and EBL fabricated nanostructures before SiO$_2$ capping; (**f-j**) Cross-sectional TEM images showing as small as <5 nm wide Si nanostructures in SiO$_2$ capping layer. Scale bar in figure (**e**) is 2 μm.

critical to reliably creating fluidic structures into a single Si layer with dimensions spanning several orders of magnitude. The fabricated nanofluidic structures (Fig. 2e and Supplementary Fig. 6) show a good alignment between DUV- and EBL-fabricated nanostructures. After nanopatterning a 40 nm-thick α-Si layer by plasma etch, we deposited TEOS-based thermal SiO$_2$ (100 nm) followed by PECVD SiO$_2$ (~2 μm) to cap the α-Si nanostructures. Cross-sectional transmission electron microscope (TEM) images show α-Si lines of different dimensions defined by EBL, from ~60 nm to as small as <5 nm wide (Fig. 2f–j).

Another unique advantage of our strategy is its potential in precise dimension tuning at Angstrom level. For example, Si nanofluidic structures can be controllably wet etched at a low rate of 25 Å per min (ref. 32), essentially allowing angstrom-level controlled reduction of the dimensions. Such a precise control at single-digit nanometre dimensions is critical to solid-state DNA sensing applications, because precise sub 5 nm manufacturing is required to linearize a single-stranded (ss) DNA molecule for sensing[33] and reliably read DNA bases[12–14] but so far remains extremely difficult to achieve using other conventional fabrication methods.

**Sacrificial Si etching.** To form the eventual nanofluidic channels, the Si sacrificial materials need to be completely extracted and then reliably sealed. This was achieved by patterning the

capping SiO$_2$ to provide access holes to the α-Si sacrificial layers (Fig. 3a,b). An important element in this step was to use small holes (300 nm) local to the nanostructures and large ports (1 mm) where the fluidic connections were made. The venting holes play a very critical role in our integration strategy to facilitate fast and efficient Si extraction. In our fluidic chip design, the fluidic ports are separated 30 mm apart to guarantee the design compatibility with single-molecule fluorescence imaging. The venting holes in our design provide >260 000 000 additional ports on a 40 mm by 40 mm chip to effectively transport the volatile etchant and byproducts during XeF$_2$ etching. Without such venting holes, it would take orders of magnitude longer time to implement the Si etching only through the fluidic ports, essentially making the process impractical.

Due to the nature of XeF$_2$ gas-phase etching, Si extraction can be carried out highly efficiently without chemically or physically damaging SiO$_2$ venting holes. The small venting hole dimensions (300 nm in diameter and ~2 μm deep, Fig. 3c,d) greatly facilitate the subsequent sealing process with PECVD SiO$_2$ deposition[34]. The high aspect-ratio of the venting holes (~7 in this work) helps minimize SiO$_2$ deposition at the nanohole bottom during sealing, thus preventing possible clogging of nanofluidic channels. The nanoholes were printed at a pitch of 1 μm in nanofluidic regions, compared to 2 μm in microfluidic regions, to further decrease the diffusion path of volatile etchant and product during XeF$_2$ etching and also ensure

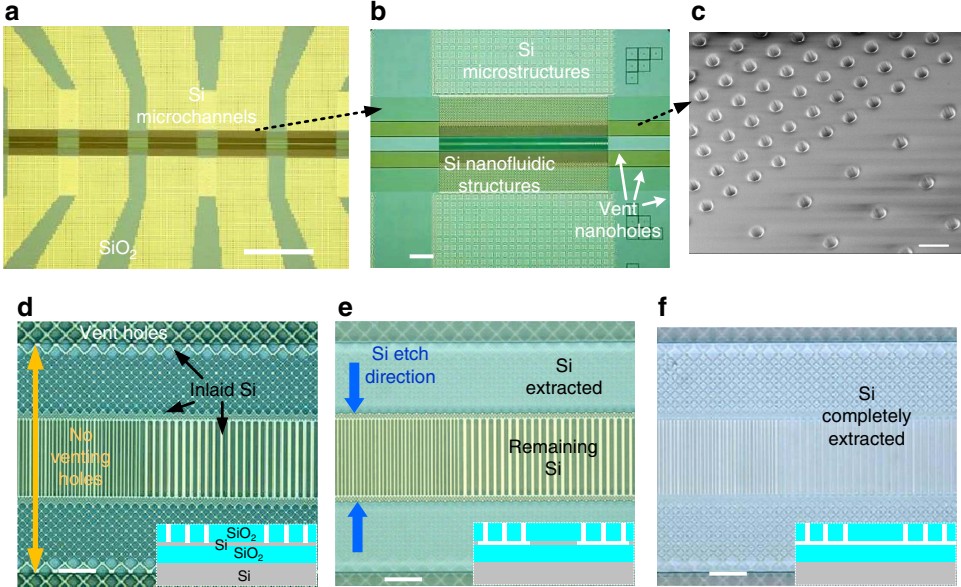

**Figure 3 | Sacrificial Si etching by XeF₂.** (**a**,**b**) Optical images of fabricated sacrificial nanofluidic device prior to Si extraction, showing: (**a**) multiple fluidic branches on one chip; (**b**) Si microchannels connected by DUV-/EBL-fabricated Si structures in the middle. (**c**) 30° tilted SEM image of venting nanoholes in SiO₂, with 300 nm diameters and different pitches (1 µm and 2 µm). (**d**–**f**) Optical images showing different stages of XeF₂ etching: (**d**) before etching; (**e**) after partial Si extraction; (**f**) after complete Si extraction. The scale bars are 1 mm, 100 µm, 1 µm in figures (**a**–**c**), respectively, and 10 µm in **d**–**f**.

landing of sufficient numbers of venting holes onto the sacrificial Si nanostructures at smaller dimensions (Supplementary Fig. 7).

Besides, the location and geometry of the venting holes can be designed to accommodate different nanofluidic structures. For example, to minimize possible damage to the critical nanostructures during venting hole formation and sealing process, we intentionally avoided nanoholes in a region spanning up to 60 µm long. Such a hole-free region was designed to consist of 20 µm long narrow channels and ∼20 µm-long two-dimensional fluidic network on each side of the channels, and can be flexibly designed and easily adjusted in DUV lithography. The large optical reflection difference between Si and SiO₂ allows reliable process monitoring during XeF₂ etching processes (Fig. 3d–f) to ensure complete removal of sacrificial Si. Importantly, we also noticed that XeF₂ can diffuse and remove sacrificial Si nanostructures in as small as 13 nm nanochannels that are not directly connected to venting holes (Supplementary Fig. 8 and Supplementary Note 4). However, the XeF₂ Si etching is slower at smaller nanostructured Si dimensions, because the vapour-phase transport of the XeF₂ precursor to Si surface and the volatile byproducts away from Si are strongly dependent on channel dimensions. Since our designed critical nanochannels (13–67 nm, Supplementary Fig. 8) are much narrower than the fluidic network connecting to them (>240 nm), XeF₂ is expected to diffuse much slower in the narrow nanochannels. Accordingly the etching time required to completely extract Si is not determined by the distance of nanochannels from the access holes but rather by the widths and lengths of narrow nanochannels. In principle, any nanochannels can be eventually extracted given long enough etching time, however, in practice specific design and manufacturing requirements may limit how long the extraction process can be and how narrow the nanochannel can be designed. Additionally, understanding that narrow nanochannels will be the time-limiting region during Si etching, we designed denser (1 µm pitch) venting holes near the nanochannels compared to in the microfluidic channel regions (2 µm pitch). Such a design

flexibility of venting holes in fact enables successful sacrificial Si extraction, making our integration strategy universal to various fluidic chip designs.

One key focus of this work is to demonstrate the compatibility of our integration strategy with complex functional two-dimensional fluidic networks (Supplementary Fig. 8). Accordingly, the mesh-like channels (∼400 µm long) are designed much longer than the straight channel portion (5–20 µm) to focus on the DNA interactions with the nanopillars. Notably, our integration strategy has no limitation to the lengths of such fluidic network and can be generalized to a variety of nanofluidic designs. Despite the fact that the etching rate drops with narrow channels, long sacrificial Si channels can be extracted by either increasing the XeF₂ etching cycles or by adding nanometre-scale venting holes on top of the channel when they are desired in design. On the other hand, narrow, long and isolated nanochannels have constrained applications because their wetting can be even more challenging than the fabrication. In this work, unnecessarily narrow and long nanochannels were not needed to achieve our desired DNA stretching in our two-dimensional fluidic network.

**Nanofluidic channel sealing**. After removal of the α-Si using XeF₂, PECVD SiO₂ was deposited to seal the small holes local to the nanochannels by 'pinch off' without restricting the fluidic ports of the device. Here, 1 µm-thick SiO₂ was deposited at the top sidewalls of the venting holes while leaving only minimal SiO₂ (∼20 nm) at the bottom (Fig. 4a,b). Hence, the sealing process did not clog sacrificial nanofluidic structures in our design (40 or 100 nm deep). The selective deposition was attributed to the fact that silane based PECVD SiO₂ growth process is rather isotropic in and around the holes, with a rate in the lateral direction only slightly slower than in the vertical, therefore the vent holes quickly close. The thickness of SiO₂ deposited at the hole bottom can be further decreased by using smaller holes and larger height/diameter aspect ratio. The amorphous nature of the PECVD SiO₂ is also beneficial for sealing as there

are no crystalline grain boundaries. In addition, this sealing process is self-terminating, because any additional $SiO_2$ deposition after pinch-off will not have any negative impact, so tight control of the $SiO_2$ thickness is not required. The high Si extraction efficiency and robust sealing process make the sacrificial approach very desirable for nanofluidic applications where the structure dimensions are in the deep nanometre scale.

The completed fluidic chips on a 200 mm Si wafer are shown in Fig. 4c with 12 fluidic chips on each 200 mm Si wafer. We

observed that our integration and fabrication strategy allowed us to consistently and reliably fabricate functional nanochannels with critical dimensions down to 20 nm on multiple 200 mm wafers with 12 nanofluidic chips per wafers. As shown in cross-sectional TEM images (Fig. 4d–h), the sealing process resulted in vacant nanochannels of dimensions down to 18 nm. Smaller sacrificial nanochannels were very challenging to image under TEM, because < 20 nm nanochannels were observed to close (for example, a 14 nm-wide nanochannel in Supplementary Fig. 9), attributed to possible electron beam induced carbon deposition[35] and/or melting of $SiO_2$ dielectric films.

To keep fluidic access ports open while reliably sealing all the venting nanoholes, we designed fluidic ports of $\sim 1$ mm in diameter connecting to the microfluidic Si structures (Supplementary Fig. 10). As a result, the 1 μm-thick sealing film would neither close the 1 mm-diameter fluidic ports nor clog the 2 μm deep microchannel, allowing fluids to be introduced into the chips.

Structural reliability is an important issue to nanofluidic devices sealed with a thin-film layer, particularly during wetting process. For example, in a slit-like fluidic channel which has a large width/height ratio (as large as 100 (refs 36,37)), it has been shown the capillary force can induce a large surface tension force to deform and even possibly collapse the sealing layer. Because the vertical mechanical deformation of the capping layer $\Delta h$ is proportional to its lateral dimension $W$, a wide and thin capping layer is subject to higher possibility of collapse or breakage, resulting in device failure. This essentially limits the lateral dimension of such nanoslits to smaller than a critical dimension. To mitigate this risk, we utilized a two-dimensional interweaved micro- and nano-scale fluidic network rather than a single wide channel for fluidic transport, and designed the lateral dimensions at nanometre scale for thin (sub 100 nm thick) channels to minimize the width/height ratio. On top of the planar nanochannels, the sealed venting nano-holes also had a small diameter/height ratio (Fig. 4a, $\sim 1.5$ μm high after sealing, $\sim 300$ nm diameter), effectively preventing the sealing $SiO_2$ layer from collapsing. Therefore, the design of nanometre scale channel and venting hole dimensions is very favourable to mechanical stability of our fluidic chip. Considering the fact that sub 10 nm channels sealed by a thin (1 μm) $SiO_2$ layer could successfully sustain

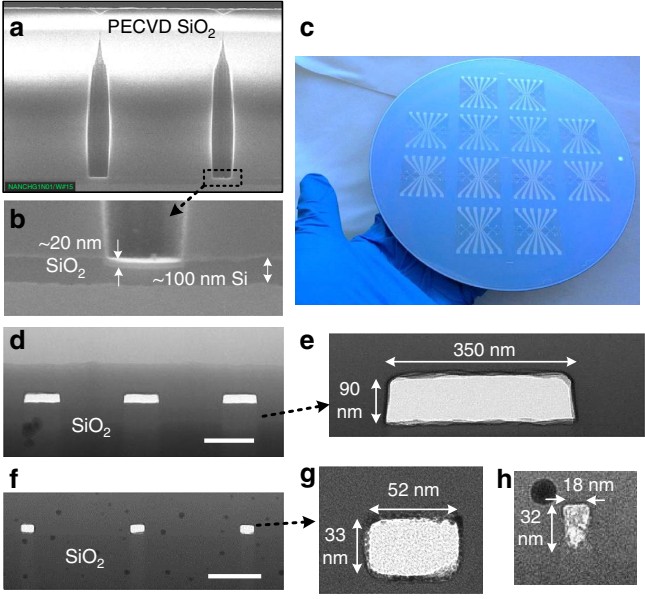

**Figure 4 | Sacrificial nanofluidic devices after sealing venting holes.**
(**a**,**b**) Cross-sectional SEM images showing venting holes after PECVD $SiO_2$ sealing, showing: (**a**) completely sealed venting holes at the top surface; (**b**) only minimal ($\sim 20$ nm) $SiO_2$ deposited at the nanohole bottom. The α-Si layer was intentionally left to visualize the deposited $SiO_2$. (**c**) Optical image of completed nanofluidic chips on an 8-inch wafer.
(**d**–**h**) Cross-sectional TEM images showing different nanochannels dimensions: (**d**,**e**) 350 nm by 90 nm; (**f**,**g**) 52 nm by 33 nm; and (**h**) 18 nm by 32 nm. The scale bars in figures **d** and **f** are 500 and 200 nm, respectively.

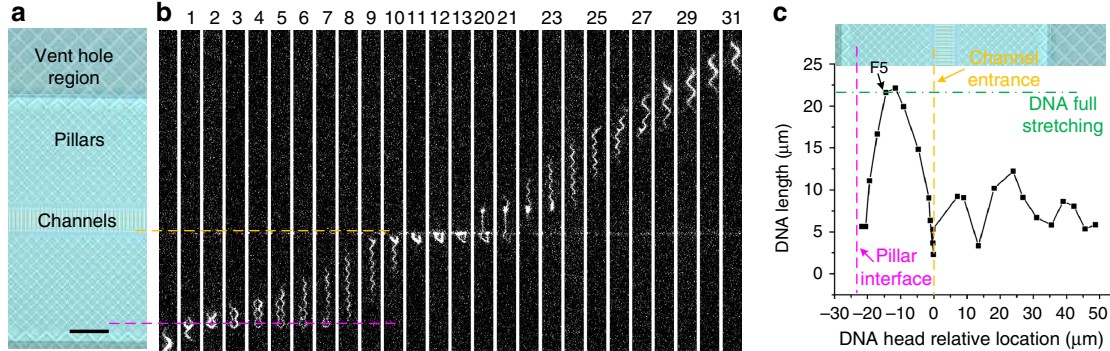

**Figure 5 | Single-molecule fluorescence imaging of DNA in sacrificial Si nanochannels.** (**a**) Optical image of nanofluidic regions with nanopillars and nanochannels (40 nm deep, 200 nm wide and 500 nm pitch). (**b**) Selected fluorescence images showing λ-DNA flowing through nanopillars and nanochannels corresponding to the optical images in **a**. Magenta and yellow dash lines indicate the pillar interface designed for straddling and the nanochannels entry point, respectively. Here frame 1 is defined the first frame the DNA molecule enters the imaged area. The DNA flowed from the bottom to the top. (**c**) The location-dependent DNA extension due to its hydrodynamic interactions with nanostructures, with the optical graph of the nanofluidic structures added as a location reference. Here the x axis origin is set as the nanochannel entry. Each black square dot represents the DNA extension in one frame, and the data point of frame 5 is labelled. The time interval between adjacent frames was $\sim 18$ ms. The horizontal green dash-dot line indicates the estimated dyed lambda DNA extension when it is fully stretched. The scale bar in **a** is 10 μm.

capillary force during DNA flow[38], we believe our sacrificial nanochannels can also reliably operate at sub 10 nm regime.

**Nanofluidic channel analysis and DNA translocation.** Using a fluorescence microscope with a customized fluidic jig (Supplementary Fig. 11), we also imaged λ-DNA hydrodynamics in the nanofluidic chip (Fig. 5). We used capillary force to load DNA into nanochannels (40 nm by 200 nm), similar to our previous work[7]. We did not observe the collapsing of nanochannels or fluidic leakage during channel wetting by either capillary force or by external pressure ($>11$ Bar), indicating a good mechanical reliability of the sealed channels. The overall mechanical strength of the sacrificial nanofluidic chip is determined by the essentially three-dimensionally structured $SiO_2$ capping and sealing layers (Supplementary Fig. 10), which function as the frame to support the nanofluidic channels and prevent mechanical breakage. To further improve the mechanical robustness, the thickness of the sealing layer could be increased (for example, from 1 μm in our experiment to 1.5–2 μm), and the area density of venting holes (currently ~300 nm wide, 1 μm and 2 μm pitch in our experiment) next to the critical channel regions can be reduced. Given the flexibility of our integration strategy to accommodate different designs, we believe this sacrificial strategy can be extended to even smaller channels.

Consecutive fluorescence images showed that DNA molecules hydrodynamically interacted with diamond-shaped nanopillars through straddling and were consequently stretched (Fig. 5), similar to our previous report using a wafer-bonded chip[7]. In our nanostructure design (Supplementary Fig. 12 and Supplementary Note 5), the hydrodynamic flow was always 45° to the nanochannels between the diamond-shaped nanopillars, thus forcing DNA to follow a zig-zag path to intentionally promote the straddling interactions. Clearly, the DNA molecule collided onto the nanopillars interface (as indicated by the magenta dash line in Fig. 5a,b), where the designed diamond-shaped pillars transition abruptly from ~3 μm wide to about ~1.5 μm and the designed pillar gaps decrease from ~700 nm to ~350 nm. Such a pillar design results in increased DNA fluidic paths and less coiled DNA[3], and as a result a higher probability of DNA straddling onto the pillars[7]. As the hydrodynamic flow moved DNA head and tail forward (Fig. 5b frames 1–4), the DNA molecule became stretched. Then as the head pulled the tail away from the pillar interface (Fig. 5b frames 5–7), a mechanical stress was applied onto the DNA, which kept the DNA in the stretched state. Our measurement results (Fig. 5c) showed that the stretched lambda DNA (48.5 kbp) reached a maximum extension of ~22 μm (frame 6), reaching its full contour length with intercalated fluorescence dyes (~30% longer than without dyes, that is, ~21.5 μm).

The demonstrated DNA stretching through the straddling interactions, with another example given in Supplementary Fig. 13, provides a promising route to linearize not only double-stranded (ds) DNA but also potentially ss DNA, because this mechanical force based method does not require the design of the nanofluidic structures smaller than the persistence length of DNA (~50 nm for ds DNA[39] and ~2 nm for ss DNA[33]), which are expensive and difficult to fabricate. Previously, we have demonstrated the straddling based DNA stretching by wafer bonding approach, and the successful integration of such functional and complex nanofluidic structures by our sacrificial integration strategy makes it possible to achieve wafer scale fabrication of single-molecule DNA sensors[7]. In addition, it is also evident that the sacrificial nanofluidic devices are completely compatible with single-molecule fluorescence imaging and very

suitable for interrogating the complex biomolecule interactions with nanostructures, making our integration strategy an ideal candidate in a variety of applications where precise, complex and large-scale integrated nanostructures are necessary for visualizing, manipulating and detecting biomolecules[10].

To further evaluate the fluidic channel structural integrity and fluidic continuity, we measured ionic conductance of fluidic channels filled with potassium chloride (KCl) electrolyte (pH 5.5) on an automated probe station (Supplementary Fig. 14 and Supplementary Note 6). The linear I–V curves at all molarities indicate good wetting and a tunable electrical conductance, which is important for electrical detection of DNA. In addition, the nanofluidic chips were capable of sustaining long-time (continuously tested $>11$ h) electrical ionic current measurements without leakage or degradation, demonstrating excellent mechanical and electric stability. Our analysis evidently demonstrated that our sacrificial Si etching strategy supports complex design of nanofluidic structures towards on-chip single-molecule studies and biomolecular manipulation.

**Discussion**

In this paper, our sacrificial Si based integration strategy is demonstrated using the existing 200 mm wafer-scale CMOS microfabrication facilities available at IBM T.J. Watson Research Center, with all the fabrication steps carefully scrutinized and proven compatible with industrial CMOS manufacturing standards. Therefore, this proof-of-concept work of functional and complex nanofluidic devices can be readily transferred to larger scale manufacturing. In our work, using integrated multi-level lithographic nanopatterning on stacked multi-layers of sacrificial Si films, we have successfully demonstrated functional nanofluidic features with their lateral and vertical dimensions spanning over orders of magnitude in a single chip (laterally from $<20$ nm to 1 mm, vertically from 40 nm to $>2$ μm). Our scalable fabrication strategy greatly facilitates complex nanofluidic system design with optimized functionalities such as fast fluidic transport and controlled biomolecular manipulation as verified by our demonstration of regulated λ-DNA straddling and stretching in nanochannels and nanopillars arrays.

The flexible integration of complex nanofluidic design, wafer-scale single-digit nanometre structure and CMOS-compatible fabrication, reliable fluidic sealing, and low thermal budget could make our strategy a universal approach to integrate functional planar nanofluidic systems with electronic circuits for lab-on-a-chip applications. Our technology holds promise in other research areas for integration of complex and precisely controlled nanomaterials and nanostructures. It is also envisioned that more complex three-dimensional structures can be integrated using our sacrificial Si strategy, by simply stacking and connecting multiple layers of sacrificial Si structures. This approach could have profound impacts across different fields, such as MEMS, nanophotonics, nanoelectronics and biosensors.

**Methods**

**Overall fabrication scheme.** Multiple levels of lithography, involving EBL (Leica VB6), DUV stepper (ASML 5500/300) and MUV printer (Suss MA8 contact mask aligner) were combined with dielectric deposition (PECVD), plasma etch and chemical mechanical polishing (CMP) steps for the fabrication. First, the alignment marks for all following lithographic levels were printed by DUV and transferred to the Si wafer substrate by plasma etch. Then, thick α-Si microstructure was patterned by MUV and plasma etch, and then planarized by CMP. Third, critical nanostructures and microstructures were fabricated in thin α-Si by three-level mixed lithography combining EBL, DUV, and MUV and plasma etch. Fourthly, nanoholes and fluidic access holes for Si venting were printed by DUV and MUV,

and then plasma-etched in the capping dielectric film. Finally, sacrificial Si was removed and venting nanoholes were sealed.

**Microfluidic structure patterning and substrate planarization.** The wafers we used for the sacrificial nanofluidic devices were 200 mm Si wafers with a $SiO_2$ layer (500 nm–1 µm). The Si microstructures were defined by MUV lithography using a 6 µm-thick photoresist (AZ 4620), and then transferred to Si by plasma etch using $HBr/Cl_2$ chemistry (Applied Materials DPS plasma etcher). The wafers were then cleaned by oxygen plasma (6 min), Piranha etch (30 min), and diluted hydrofluoric acid etch (100:1, 15 s) to strip the photoresist and remove surface contaminants. Twelve microfluidic chips were designed on a 200 mm wafer, each chip of 40 mm by 40 mm square, and each chip hosted six independent microfluidic channels for multiplexed sample testing. In our design, the microstructures were 10 µm wide and ∼2 µm deep.

Then a $SiO_2$ layer was conformally deposited on the micropatterned α-Si structures by PECVD (Applied materials Hex 543A). The desired thickness was achieved in multiple steps, each step depositing up to 1 µm. Ellipsometry was utilized to calibrate the deposition rate and monitor the film thickness. The CMP step was performed (Ebara Frex200 Polisher) to minimize step heights across the α-Si structures. In our experiment, the coated $SiO_2$ thickness was chosen to be slightly (500 nm–1 µm) thicker than the sacrificial Si thickness to ensure an optimized planarization without lengthy polishing. The polishing process was separated into two stages using a conventional silica slurry with sufficient $SiO_2$ polishing rate to handle the thick deposition. The first step polish was defined with a parameter set to provide an acceptable bulk removal polishing rate ($1.5$–$2.0$ nm s$^{-1}$) to remove 80–90% of the $SiO_2$ layer on top of Si microstructures with minimal loss in the surrounding field, thus minimizing the topography to be planarized in next step. This stage was followed by a second step polish with a parameter set designed to slow the polishing to a much slower touch-up rate ($\sim 0.5$–$1.0$ nm s$^{-1}$) to expose the inlaid Si without the introduction of a significant step height between the Si and surrounding $SiO_2$ while minimizing removal (loss) of the exposed inlaid Si materials. With a less than optimal $SiO_2/Si$ polish selectivity for the slurry and to counter limitations on within-wafer non-uniformity, the second stage polishing was further divided into multiple steps with each consisting of a short CMP duration (10–20 s). In addition, both film thickness and optical microscope inspections were performed between the short CMP steps to further ensure complete uniform clearing across the wafers.

**Fabrication of sacrificial nanofluidic structures.** The detailed fabrication steps of sacrificial nanostructures are sketched in Supplementary Fig. 5. Briefly, we first coated the substrate with an OPL (JSRMicro, USA) of 65 nm. Then 1% HSQ was spin coated at 3,000 r.p.m. without baking to yield a thickness of ∼20 nm. The HSQ was exposed on a Leica EBL system at a dose between 1,800–2,700 (µC cm$^{-2}$), and cured (8 torr under 4,500 sccm $N_2$ flow, 400Ω, 300 s). Then 450 nm-thick DUV resist (UV 110, Dow Chemical Company) was spin-coated, soft baked at 90 °C for 60 s, printed by a DUV stepper (ASML 5500/300), post-exposure baked at 100 °C for 60 s, and developed in standard 0.26 N tetramethyl ammonium hydroxide developer.

After EBL/DUV lithography, plasma etch was performed to transfer the pattern through an OPL and HM layer. The pattern was etched through OPL using an $NH_3$ gas discharge in an Applied Materials Enabler plasma etcher. Then, the HM was patterned using a $CHF_3/CF_4$ gas discharge in an Applied Materials DPS plasma etcher. The left-over OPL layer was stripped using an Applied Materials Axiom asher and wet cleaned by dilute HF and sulphuric/nitric acid mix (10 min). The wet cleaning step was used to completely clean away resist and OPL, in case plasma etch could harden organic surfaces that might be left as residue after plasma strip.

Then, 1 µm photoresist (TOK, THMR-iP3250) was spin-coated, exposed and developed to align the microstructures to the EBL/DUV features in HM. After MUV lithography, plasma etch was performed to transfer the pattern through the HM layer using a $CHF_3/CF_4$ gas discharge in an Applied Materials DPS plasma etcher. The MUV resist was then removed by plasma strip using an Applied Materials Axiom asher. Then the patterns in the HM were transferred into the thin α-Si layer using $HBr/O_2$ plasma etch in the DPS etcher. A wet cleaning (sulfuric and nitric acids, 10 min) was used to further ensure complete stripping of residue photoresist. The remaining thin $SiO_2$ HM would not affect following processes and did not have to be stripped.

**Sacrificial Si extraction.** First, the fabricated Si wafers were subjected to a 100 nm $SiO_2$ deposition by a thermal TEOS process (precursor Tetraethyl orthosilicate, 400 °C, 60 torr) and then 2 µm $SiO_2$ by PECVD (Applied materials Hex 543A). Then the wafers were subjected to a touch-polish to reduce the surface topography prior to DUV printing nanoholes for Si extraction. The surface topography was reduced to 16 and 8 nm, respectively, from patterned wafers with originally 100 and 40 nm-thick α-Si layers. Then, to pattern nano-holes for venting, HDMS (Hexamethyldisilazane) and 825 nm-thick DUV resist (UVII, Dow Chemical Company) was spin-coated. The resist was soft baked at 90 °C for 60 s, printed by a DUV stepper (ASML 5500/300), post-exposure baked at 100 °C for 60 s, and developed in standard 0.26 N tetramethyl ammonium hydroxide developer for 50 s.

The printed nanoholes were etched by plasma etch (Lam 4520 XL) to etch through the deposited $SiO_2$. Then the resist was stripped by oxygen plasma. To keep fluidic ports open after sealing the nanoholes, we printed large fluidic access ports (∼1 mm diameter) in 1 µm thick photoresist (TOK, THMR-iP3250) by MUV lithography, and transferred the structure to the $SiO_2$ layer by plasma etch.

Before $XeF_2$ etching, the substrate was cleaned by HF dip etch (600:1 diluted hydrofluoric acid, 30 s) to remove native oxide to make sure that all the exposed Si began etching at the same time. The $XeF_2$ etching was carried out in a Xactic etching chamber, which used $XeF_2$ mixed with $N_2$ to etch exposed Si sample in a cyclical pulse mode (3 Torr $XeF_2$, 15 Torr $N_2$, one pulse time of 10 s). After each etching cycle, the chamber was evacuated and the process was repeated until all the sacrificial Si material was removed. The $XeF_2$ removed the thick and thin α-Si layers simultaneously through the small nanoholes and fluidic ports. Experimentally, the last regions to clear were the nanochannels, where the $XeF_2$ access to the Si was based on diffusion from adjacent nanoholes and limited to the cross-sectional area of the nanochannels. The required number of pulses to completely remove Si was highly dependent on the nanochannel design and was about 90 pulses in this work.

**DNA translocation and fluorescence imaging.** A fluidic jig was customized for DNA fluorescence imaging, consisting of three components: a mounting base for leveling and positioning the chip, a flow chamber with fluidic access ports for sample handling and a pump connector to provide a vacuum driving force. The flow chamber was designed with a rectangular opening in its centre to accommodate ×100 oil-immersion objectives for fluorescently imaging the DNA in all the critical nanofluidic regions. The customized fluidic system allowed us to simultaneously image the DNA translocation and control the hydrodynamic or electrophoretic flow.

The stock solutions of λ-DNA (48.5 kbp, 0.5 µg µl$^{-1}$, New England Biolabs) were sequentially diluted to the use concentration (10–100 pg µl$^{-1}$) using 10 mM TE buffer (10 mM Tris , 1 mM EDTA, PH 8, Life Technologies) mixed with 3% oxygen-scavenging 2-mercaptoethanol (Sigma-Aldrich Co.) and 0.1% TWEEN 20 (Sigma-Aldrich Co.). The diluted DNA was labelled with fluorescence dye YOYO-1 Iodide (491–509; Life Technologies) with a DNA bp to dye ratio of 5:1, incubated at room temperature for 2 h, and stored at 4 °C for use. For DNA fluorescence imaging, 40 mm by 40 mm chips were diced off from the manufactured 200-mm wafers, and connected to the fluidic jig. The prepared DNA sample in TE buffer was directly loaded into an empty channel, and driven by capillary force to travel from the reservoir to the channels. Cautions were taken to avoid generating bubbles in the jig reservoirs. The fluidic jig was mounted onto an upright Zeiss microscope (Axio Scope.A1, Zeiss), and the fluorescent DNA signals were excited by a 470 nm blue LED and collected by a CCD camera (Andor iXon Ultra 897) through a ×100 oil-immersion objective with a numerical aperture (NA) of 1.25. The excitation filter, beam splitter, and the emission filter were 470/40 nm, 495 nm, and 525/50 nm, respectively.

**Data availability.** The authors declare that the data supporting the findings of this study are available from the authors and presented within the article and its Supplementary Information file.

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

## Acknowledgements

We acknowledge many fruitful discussions with Roche 454 Life Sciences colleagues and financial support for the work from Roche Applied Sciences. We are grateful to H. Peng, D. Wang, J. Bai, Y. Astier, J.T. Smith, B. Luan, K. Reuter, S.M. Rossnagel, M. Guillorn, W.H. Advocate, C.M. Breslin, J. Bucchignano, E.A. Duch, A. Galan, E. Kratschmer, S. Meshram, P.J. Litwinowicz, W. Price, M.A. Pereira, J. Mailing, A. Stamper, J. Hedrick, R. D. Goldblatt, D. Pfeiffer, A. Royyuru, T.C. Chen and the IBM Microelectronics Research Laboratory (MRL) staff for their discussions and contributions to this work. These devices were fabricated in the MRL at the IBM T. J. Watson Research Center using wafers provided from the IBM Burlington Vermont facility. C.W. appreciates S. Lindsay at Arizona State University for reading the manuscript.

## Author contributions

C.W. wrote the manuscript with comments and edits from all authors. G.S. and Q.L. directed the overall research project. S.P. proposed CMOS-compatible nanofluidic integration strategy. C.W., C.V.J. and E.G.C. designed the sacrificial process integration scheme. C.W., E.G.C. and S.P. designed the mask set. S.-W.N., J.M.C., and R.L.B. coordinated the fabrication. M.B., M.F.L, J.V.P., C.V.J., E.A.J., S.P.R., and Q.L. contributed critically to lithography and oversaw the wafer-scale fabrication work. L.M.G. performed TEM analysis. C.W. and G.S. prepared the DNA sample, built the fluidic jig and fluorescence imaging set-up, and performed the fluorescence DNA imaging and analysis. S.P. and E.A.J. designed waver-scale electro-fluidic robotic test station. C.W., S.-W.N. and S.P. performed electrical measurement. All authors contributed to data analysis and interpretation.

## Additional information

**Competing financial interests:** The authors declare no competing financial interests.

