## [Peer Review File · Nature Communications]

Reviewers' comments:

Reviewer #1 (Remarks to the Author):

I have reviewed Wang et al., which is reporting a wafer scale CMOS compatible fabrication process of a few nanometer nanochannel fabrication process. Nanofluidic systems are emerging fluidic MEMS devices with many potential future applications in biological and non-biological applications. There are significant level of academic research ongoing, yet reliable manufacturing of such devices have never been demonstrated before. My impression is that this is a substantial and meaningful work to be published in Nature Communication, mainly because reliable, manufacturing-scale fabrication of these novel devices would significantly facilitate translation of nanofluidic systems into commercialization. However, there are many critical issues that make the validity of the paper in question (see below). I would like to ask the authors to address the issues listed below before being considered for possible publication in Nature Communication.

1. The main claim of this paper is that the authors implemented the fabrication of nanofluidic channels in a large area wafer scale, and with methodologies that are compatible with standard CMOS process. This is, if validated, not an insignificant milestone given how much success CMOS process has been making in advancing microelectronics. Yet, this paper does not discuss the critical issues related to the reliability of the 200nm wafer scale fabrication, such as

- Uniformity of the nanochannel fabricated across the wafer (single digit nanochannel)
- Wafer-to-wafer variation
- Overall yield of the fabrication

2. XeF₂ etching is not a new technique in MEMS, and this has been previously used for etch out sacrificial layers to build a microfluidic channel. The authors should clearly list relevant references regarding the use of the etching technique for building microchannels. What is novel about this work? In addition, it appears that the existence of "venting holes" is critical to the success of the fabrication. What happens when there are no venting holes, or when venting holes are far away from the nanochannel? This requirement will surely introduce some significant constraint in overall design of the system. How far the venting holes can be? What is the maximum length of the nanochannel one can build with reasonably placed venting holes?

3. The last part of the paper is simple experimental demonstration of the proper nanochannel function, by showing DNA stretching. But this part is simply too limited to ensure that one actually has the nanochannel that are in many ways similar to what has been reported. The DNA stretching achieved in this paper is too limited and we do not see the evidence of full DNA stretching. In addition, it is well known that nanochannel conductance will plateau (due to surface conductance domination) when the salinity of the water decreases below a certain limit, which is widely accepted as the hallmark of nanochannel's unique electrical properties. The authors should demonstrate this to show that the nanochannel fabricated here is of the same properties as others. Another clear tell-tale signature is the generation of ion concentration polarization (ion depletion) in moderate buffer concentrations.

4. One of the challenges of the nanochannels, especially the one built by sacrificial layer etching with thin capping layer, is that the filling of very small nanochannel can induce capillary-driven negative pressures within the nanochannel during the filling process, collapsing the nanochannel. Do authors observe this phenomenon? Is this related to the fact that the authors are showing actual DNA data from relatively large (~40nm) channels, not from very small nanochannels of ~10nm or less dimension?

5. While the DNA stretching and manipulation was probably the first application of nanofluidic devices

(circa ~2000), it is now widely considered as an alternative membranes (with controlled nanopores). In this regard, it would be important to have not just the accurate nanochannels, but also many of them in parallel, to achieve as a high throughput as typical membranes (nanofiltration membranes, for example). I wonder whether this process can allow stacking more nanofilter layers on top of each other, or other means of generating massively parallel nanochannels.

6. Lastly, I think authors should discuss one critical issue in this paper, which is the overall cost of the nanochannel fabrication. The lithography and other CMOS compatible processes used in this work requires expensive, unique tools of fabrication, achieving high spatial resolution. It will be expected that the cost of fabrication would be relatively high, although the cost can be reduced in the context of large volume manufacturing, as one can see that such a fabrication can be done economically in volume production. (USB dongles with many gigabytes of flash memory are now much less than 10\$.) I believe that the authors should discuss the overall economics of building the device at different volume scales, since the focus of the paper is on the manufacturing-level fabrication of nanochannel. I think it would be very expensive to make 1 wafers with nanochannels. How many wafers (~1000? ~1,000,000 wafers?) one needs to make to achieve the reasonable cost efficiency of the process?

Overall, I believe this is a potentially very impactful paper, but the authors are falling far short of truly convincing the readers that nanochannels can now be made as reliably as typical microelectronics devices. I would hope to see authors to fill this gap and make this work true to its original claims.

Reviewer #2 (Remarks to the Author):

Wang et al. in this manuscript present an approach to fabricate nanochannels with critical dimension less than 20 nm at wafer-scale (200 mm) and then investigate its application in single DNA translocation analysis. The fabrication strategy is consistent with standard semiconductor processing, which presents a number of advantages: (1) The monolithic nanofluidic chip integrates microchannels and nanochannels together, which both improves the efficiency of fluidic transport and realizes well-controlled DNA translocation; (2) Robust sealing processes for multilayered nanostructure avoid selective sealing or wafer bonding steps, which means this nanofluidic system can operate at higher pressures and longer times than other designs; (3) Using silicon as a sacrificial material for microchannels and nanochannels avoids wet etching, which often requires longer times and makes it difficult to precisely control the final size and shape of nanochannels; (4) Because it employs a planar (horizontal) fluidic structure rather than a vertical nanopore geometry for studying the single DNA translocation, this design is well-suited to directly observing the motion of DNA molecules in confined nanostructures by fluorescence. Thus, the approach has a lot going for it, and the paper is well-written.

My reservation concerns the fact that the fabrication processes, the layout of the nanofluidic device, and the results of single molecule detection are very similar to previously published work, which is not cited in the manuscript. (IEEE, IEDM13-369-372, 2013 conference). For example, a number of figures in this manuscript, *e.g.* Fig. S3(c) vs. Fig 3(A); Fig. S8 vs. Fig. 6; Fig. 2(f)-(j) vs. Fig. 8; Fig. 4(a) vs. Fig. 7(B); Fig. 4(d) vs. Fig. 9(A); Fig. 4(e) vs. Fig. 9(c); Fig. 4(g) vs. Fig. 9(D), and the i-V curves in both papers are almost the same as the previous paper. The numerous similarities raise the issue that the present manuscript is derivative of the earlier conference publication. Considering that *Nature Communications* requires papers with "important advances of significance to specialists within each field", the authors need to provide a more detailed explanation of the details of this research which speak to its novelty when compared with their previous work. In the absence of such an explanation, I could not recommend publication of the manuscript in *Nature Communications* in its current form.

Reviewer #3 (Remarks to the Author):

The authors present a wafer-scale fabrication process for producing nanofluidic chips with single-digit nanometer dimensions compatible with standard CMOS processes. The advantages claimed include wafer-scale manufacturing, reliable sealing, and a low temperature process allowing for integration with CMOS circuitry for lab on a chip applications. The presented work is distinguished by the novelty of their process, specifically using sacrificial silicon structures etched by Xenon difluoride gas. The experimental results, observing DNA stretching and translocation, look promising and represent an important potential use of the device.

The advantages claimed seem logical in respect to the current field of research and the work is very well communicated. Overall, the field of micro- and nano-fluidics is very active and this work makes strides in advancing the processes and procedures used.

Specific comment:

A large claim of the paper is the wafer-scale capability of the fabrication and its compatibility with traditional CMOS. As mentioned in the paper, yield is a large and important issue in developing CMOS-compatible fluidic structures, yet data (channel conductance measurement) from only one channel from one device is presented. The claims of the paper would be greatly enhanced with data from more than one channel, and preferably, across the devices on the manufactured wafer. If the yield is poor, it would be beneficial for the reader to understand where the process needs further improvement.

Reviewers' comments:

Reviewer #1 (Remarks to the Author):

The authors have responded to the reviewer's criticisms in the revised paper. But I still have one major issue, plus several minor points about this paper, which prevents me from supporting the publication of this paper.

Major Issue: The authors' response to my comment (as well as the reviewer #3) on the issue of yield, uniformity, and variation is unacceptable. Simply put, we do not have any information from this paper regarding the yield, uniformity, reliability of THIS SPECIFIC PROCESS, and the authors are providing generic arguments about the reliability of established CMOS process in general. From this paper, I wonder if;

1. The authors have so far created just one successfully fabricated chip without any defect, out of many 200mm wafers attempted.
2. The authors have so far created one successful wafer, with good uniformity across the entire wafer that was successful, but with significant wafer-to-wafer variations.
3. The authors have established many successfully fabricated wafers with the same uniformity and quality across the board.

What is the case in this work? The authors simply claims the great uniformity of the CMOS process, which I agree. But they do not provide any information regarding the yield of this specific devices and processes. In typical reviews, we tend not to focus too much on this issue of yield, but in this paper, this information is critical. This is because, as the authors are pointing out as well, all of the scientific concepts have been already known and published before, including nanochannels and their DNA stretching based sensing, nanochannel fabrication by XeF₂ sacrificial etching, and even the use of access holes (followed by cover-up/resealing) to facilitate the removal of sacrificial layer (see S. W. Turner et al., J. Vac. Sci. Technol., B 16, 3835 (1998).). The only major/novel claim is the wafer-scale integration of these tricky processes, but I don't see any evidence of such "wafer-scale integration" in this work. If the yield is limited, then (as the third reviewer points out) the authors at least should try to point out the reasons, to really advance the field as typical Nature-level papers do.

Minor issue

In response to my comments, the authors greatly expanded the discussion on the design of venting hole, and how one can even fabricate very small nanochannels by properly locating the holes on a non-nanochannel section. They also argue that their focus is on short, one-dimensional nanochannel fabrication for the specific goal of DNA stretching. I would like to remind the authors again that the main claim of this work is the process, not the particular chip (if it were only about linear nanochannel fabrication, the impact of the work would be diminished). I would encourage authors to add more design-limitation discussions, to help the readers when this process can or cannot be used, and how. Specifically, what is the practical limitation of nanochannel width vs. distance of nanochannel from access hole vs. etching time allowed (given the etch selectivity)? Such discussions are essential for a paper claiming to report novel/innovative processes. It is probably OK for authors not being able to make extremely long and narrow nanochannels, but I still think it is authors' obligation to clearly tell what can and cannot be made using their process.

Reviewer #2 (Remarks to the Author):

The authors have submitted a revised manuscript which addresses a number of issues. However, they have not cited the previous conference proceeding identified in the original review - this despite their own persuasive arguments that the earlier proceedings paper does not compromise the novelty or need/ability to publish in a full paper. Although this is a strong paper in the main, in this point, at least, the revision is not responsive to the review.

Reviewer #3 (Remarks to the Author):

The authors have comprehensively and constructively addressed all of my previous comments. I would like to recommend publication of this paper in Nature Communications.

REVIEWERS' COMMENTS:

Reviewer #1 (Remarks to the Author):

I have reviewed the second revision of Wang et al. and, after a long time debating with myself, my conclusion is that this paper, at least in my opinion, is not representing the advances and breakthroughs we typically associate with journals like Nature Communication. On one hand, I do think strongly that more emphasis and credit should be given to the technological breakthrough (not just scientific outcomes), since those who are working on the technology side of things tend not to get the justified credit. But, on the other hand, in my judgement, many authors of the previous papers cited in this paper (related to the earlier implementation of the fabrication technology described in this paper) will find this work to be rather incremental and not really advancing things far ahead from what they have achieved in the past.

1. Based on the responses, it appears that the authors did not reach the level of "reliability testing" that are often associated with the industrially implemented process. They are offering to change the title from "manufacturing" to "fabrication". But, "fabrication" of nanochannels described in this paper has been achieved using other techniques. There is no doubt that new ideas are used in this work, but I don't think the level of innovation described in this paper can be called "transformative" or "really advancing the field".

2. The fabrication of nanochannel demonstrated in this work is impressive, but not drastically different from earlier implementation for DNA stretching / mapping applications. For those applications, you only need ~50micron or less long 1D nanochannels, and the authors are demonstrating the techniques (how to design access holes, etc.) for both 1D nano channel and the loading and unloading channels (this can be considered as 2D nanochannel). In my judgement, this was an (important but not necessarily ground-breaking) extension of previous sacrificial etching based nanofilter fabrications papers.

Reviewer #1 (Remarks to the Author):

I have reviewed Wang et al., which is reporting a wafer scale CMOS compatible fabrication process of a few nanometer nanochannel fabrication process. Nanofluidic systems are emerging fluidic MEMS devices with many potential future applications in biological and non-biological applications. There are significant level of academic research ongoing, yet reliable manufacturing of such devices have never been demonstrated before. My impression is that this is a substantial and meaningful work to be published in Nature Communication, mainly because reliable, manufacturing-scale fabrication of these novel devices would significantly facilitate translation of nanofluidic systems into commercialization. However, there are many critical issues that make the validity of the paper in question (see below). I would like to ask the authors to address the issues listed below before being considered for possible publication in Nature Communication.

1. The main claim of this paper is that the authors implemented the fabrication of nanofluidic channels in a large area wafer scale, and with methodologies that are compatible with standard CMOS process. This is, if validated, not an insignificant milestone given how much success CMOS process has been making in advancing microelectronics. Yet, this paper does not discuss the critical issues related to the reliability of the 200nm wafer scale fabrication, such as

a. Uniformity of the nanochannel fabricated across the wafer (single digit nanochannel)

b. Wafer-to-wafer variation

c. Overall yield of the fabrication

We appreciate the reviewer's comments. We agree with the referee that reliable manufacturing of such nanofluidic devices have never been demonstrated before. Manufacturing-scale fabrication of these novel devices would significantly facilitate translation of nanofluidic systems into commercialization. This is precisely what we aimed to demonstrate in this work. Our nanofluidic chip design and fabrication processes on 200 mm wafer-scale platform could be directly transferred to a chip foundry for production. We adopted a CMOS compatible fabrication infrastructure, the Microelectronic Research Laboratories (MRL) at the IBM Thomas J. Watson Research Center, to fabricate our nanofluidic chips. The IBM MRL fabrication infrastructure includes Class 100 cleanroom with 200 mm wafer-scale film (metal, semiconductor and dielectric) deposition, 193 nm-, 248 nm- and MUV- optical lithography, e-beam lithography, Reactive Ion Etching, Chemical-Mechanical-Polishing, as well as all relevant metrology and testing tools. The IBM MRL is the birthplace of many groundbreaking innovations in the IC industry, including bipolar technology, SiGe, SOI, Cu interconnects, CMP, and many generations of CMOS technology for several decades. For example, we also demonstrated that IBM MRL is capable of achieving >90% electrical yields consistently from wafer to wafer, see the following figure.[1]

Fig. 4. (Color online) Electrical yields of 45 nm node fatwire level of Cu/248 nm photo-patternable low- κ BEOL structure. These yields were obtained from three line/space test macros, called BEOLCAP macros (BCLP, BLCP_A, and BLCP_B). The nominal feature size of these test macros is 320 nm.

We agree that within-wafer and wafer-to-wafer critical dimension uniformity are key metric for manufacturing-worthy fabrication process. We have added the following to our manuscript (page 6) and supplementary document (section 3.2) to discuss the nanopatterning capability of our integration strategy.

“In this work, we utilize established procedures and recipes at IBM MRL lab during critical nanopatterning steps to maximize the feature uniformity and yield. For example, the plasma etch uniformity in our etch chamber has an etch rate uniformity of within 5% across a 200mm wafer and from wafer-to-wafer. The critical dimension (CD) in DUV lithography has a <15nm variation for a 200nm line/space standard design across a 200 mm wafer, and the yield is about 100% for the dimensions in this work (critical dimension ~200 nm). The high yield is achieved by printing in a controlled and fully automated environment of an ASML and TEL track without manual handling and by applying internal stepper diagnostics on a regular basis to control the focus and dose.”

“In EBL, established CMOS fabrication infrastructure at IBM T.J. Watson research center was used to achieve high resolution and repeatable nanopatterning.[1] To achieve a good control of feature dimensions and maximize the process yields, monitor wafers were added to each batch and processed together with device wafers. This approach enabled us to optimize the recipes for the nanostructure patterning and minimize batch-to-batch and wafer-to-wafer variations. Selected SEM images of the EBL-patterned nanostructures are shown in supplementary Figure S6.”

Besides manufacturing of fluidic structures at nanometer scale, we would like to emphasize that one key advantage of our sacrificial Si based integration strategy is its capability of integrating complex and functional nanofluidic systems of various dimensions and shapes to fulfill the design requirements of different nanofluidic devices. This has already been discussed in the introduction paragraph of our manuscript, as listed below:

“However, unlike complementary metal-oxide semiconductor (CMOS) chips which utilize a relatively small range of feature dimensions, nanofluidic chips must integrate more sophisticated three-dimensional architectures incorporating vacant and sealed nanostructures with dimensions spanning several orders of magnitude to optimally detect and manipulate biomolecules. The stringent requirement of reliably forming and sealing complex and small nanostructures makes it very challenging to manufacture nanofluidic chips over a wafer scale by current CMOS processes, and thus has seriously hindered electronics integration.”

We also modified our abstract as follows:

“Here we report a scalable 200 mm wafer-scale fabrication strategy capable of producing nanofluidic chips with complex designs and down to single-digit nanometer dimensions. Compatible with industry standard complementary metal-oxide semiconductor (CMOS) logic circuit fabrication processes, this strategy extracts a patterned sacrificial silicon layer through hundreds of millions of nanoscale vent holes on each chip by gas-phase XeF_2 etching, hence dramatically improving the Si extraction yield for large-scale manufacturing. After sealing the vent holes by thin film deposition, the chips can also sustain high pressure (>11 Bar) and long-time operation (>11 h). Verified by single-molecule fluorescence imaging, we demonstrate these sacrificial nanofluidic chips can function to controllably and completely stretch lambda DNA in a two-dimensional nanofluidic network comprising channels and pillars.”

2. a. XeF_2 etching is not a new technique in MEMS, and this has been previously used for etch out sacrificial layers to build a microfluidic channel. The authors should clearly list relevant references regarding the use of the etching technique for building microchannels. What is novel about this work?

b. In addition, it appears that the existence of "venting holes" is critical to the success of the fabrication. What happens when there are no venting holes, or when venting holes are far away from the nanochannel? This requirement will surely introduce some significant constraint in overall design of the system. How far the venting holes can be?

c. What is the maximum length of the nanochannel one can build with reasonably placed venting holes?

We appreciate the comments by the reviewer.

a. We agree with the referee that XeF_2 etching of Si is an established technique that has shown its compatibility with Si processing techniques. For exactly this reason, we chose this method for wafer-scale integration. We have searched literature regarding XeF_2 related nanofluidics with the best efforts, and our findings confirm that our work differs significantly from others in many aspects. The following has been incorporated into the manuscript (page 4) to address the novelty of our integration strategy.

“ XeF_2 etching of Si is a well-established technique with demonstrated compatibility with Si processing. [2, 3] However, to date, its applications in functional nanofluidics have been rather limited.[4] Our demonstration significantly differs from others XeF_2 based integration methods. First, previous demonstrations usually have large lateral dimensions, e.g. from 10 to 100 μm [4]. In contrast, the critical

dimensions of our devices are about 3 orders of magnitude smaller. Second, single-molecule imaging and manipulation (e.g. DNA stretching), which is important to biomolecular sensing, sorting, etc., has not been demonstrated previously in sacrificial nanochannels. Here we visualize fluorescently labelled single DNA molecule flow and verify their controlled stretching in nanochannels. Third, previous XeF_2 Si etching was only applied to simple geometries such as straight and long channels but not complex and functional fluidic network. In this work, we prototype rationally designed two-dimensional fluidic structures featuring controlled DNA hydrodynamics and stretching. Fourth, conventional methods diffuse XeF_2 only through the fluidic ports to extract Si, which is inherently a time-consuming diffusion-limited process and strongly dependent on the channel dimensions. Differently, we rationally integrate venting holes to initiate the Si exaction in a parallel fashion, hence significantly reducing the manufacturing time, increasing the throughput, and enabling complex fluidic design over large areas. Lastly, we integrate our process completely on a 200 mm Si processing platform, proving the compatibility of our integration strategy with large-volume production.”

b. We believe the addition of venting holes allows much faster Si etching and supports designs of more complex structures and nanostructures. We have added the following sections to the manuscript (page 8) to better explain the venting hole issues.

“The venting holes play a very critical role in our integration strategy to facilitate fast and efficient Si extraction. In our fluidic chip design, the fluidic ports are separated 30 mm apart to guarantee the design compatibility with single-molecule fluorescence imaging. The venting holes in our design provide >260,000,000 additional ports on a 40mm chip to effectively transport the volatile etchant and product during XeF_2 etching. Without such venting holes, it would take orders of magnitude longer time to implement the Si etching only through the fluidic ports, essentially making the process impractical.”

“Besides, the location and geometry of the venting holes can be designed to accommodate different nanofluidic structures. For example, to avoid possible damage to the critical nanostructures during venting hole formation and sealing process, we intentionally avoided nanoholes in a region spanning 60 μm long, and the coverage area of the hole-free regions can be flexibly designed and easily adjusted in DUV lithography. In our experiment, we found the XeF_2 can in fact diffuse and remove the Si nanostructures of as small as 13 nm in the hole-free regions, with the etching rate increasing monolithically with channel width (supplementary Figure S8). Additionally, understanding that narrow nanochannels will be the time-limiting region during Si etching, we designed denser (1 μm pitch) venting holes near the nanochannels compared to in the microfluidic channel regions (2 μm pitch). Such a design flexibility of venting holes in fact enables successful sacrificial Si extraction, making our integration strategy universal to various fluidic chip designs.”

We also have added more details of etching nanometer scale channels to the supplementary document (Section 4.2), as follows.

“In our layout-design, the venting holes are separated by 60 μm (Figure 3). The sacrificial Si materials in the nanochannels did not have any venting holes patterned on the top; instead, they are extracted through the venting holes patterned on top of the micrometer- and nanometer-sized channels

connecting the critical nanochannels. To investigate the effect of the feature size on the XeF_2 etching process, we monitored the location of the Si etch-front by optical microscopy (Figure S8 a). In the experiments, we tested the feature size from ~ 70 nm down to ~ 13 nm. Clearly, the XeF_2 etching rate of amorphous Si in wider nanochannels was much faster than narrower channels (Figure S8 b)."

"The size-dependent Si etching rate can be understood as a result of size-dependent vapor-phase transport of the XeF_2 precursor to Si surface and the volatile byproducts away from the Si surface. Obviously, the diffusion of XeF_2 gas and by-product is slower within narrower channels. This can be attributed to higher probability of gas molecules to collide with the nanochannels sidewalls at vacuum (3 Torr XeF_2 , 15 Torr N_2 in our experiment), in agreement with Knudsen diffusion model. Our experimental results also showed a linear dependence of etching rate versus channel dimensions, probably because the diffusivity is proportional to the critical dimensions of the nanochannels at the Knudsen diffusion regime. In spite of the slow etch-rate in the narrow (sub-20 nm) nanochannels, the successful etching can be completed by increasing XeF_2 gas-purging time and cycles. In our experiments, 20 μm long, sub-20 nm wide, and 40 nm high nanochannels were successfully extracted."

Figure S8 Dimension-dependent Si extraction process in nanochannels. (a) An optical microscope image showing the evolution of etch-front of the Si nanochannels during XeF_2 etching process. The widths of the nanochannels are 13 nm, 16 nm, 18 nm, 31 nm and 67 nm. The Si thickness is 40 nm. (b) The etch front plotted against the channel-widths (right), showing the feasibility of XeF_2 Si etching at sub-20 nm.

c. From the authors' research experience with nanofluidics,[5, 6] the channel length is not necessarily the figure of merit to evaluate a design or fabrication process, because different fluidic applications require drastically different design parameters to achieve the desired performance. For example, in our experiment, the structure design and dimensions of nanopillars are very important for DNA stretching.[5] On the other hand, the fabrication of a long nanochannel using our integration strategy is completely feasible. The following has been added to the manuscript to address the concern regarding the channel length:

“One key focus of this work is to demonstrate the compatibility of our integration strategy with complex functional two-dimensional fluidic networks (supplementary Figure S8). Accordingly, the mesh-like channels (~400 μm long) are designed much longer than the straight channel portion (5 to 20 μm) to focus on the DNA interactions with the nanopillars. Notably, our integration strategy has no limitation to channel lengths and can be generalized to a variety of nanofluidic designs. Despite the fact that the etching rate drops with narrow channels, long sacrificial Si channels can be extracted by either increasing the XeF₂ etching cycles or by adding nanometer-scale venting holes on top of the channel when they are desired in design. On the other hand, narrow, long, and isolated nanochannels have constrained applications because their wetting can be even more challenging than the fabrication. In this work, unnecessarily narrow and long nanochannels were not needed in our two-dimensional fluidic network and irrelevant to the novelty of our integration strategy.”

3. a. The last part of the paper is simple experimental demonstration of the proper nanochannel function, by showing DNA stretching. But this part is simply too limited to ensure that one actually has the nanochannel that are in many ways similar to what has been reported. The DNA stretching achieved in this paper is too limited and we do not see the evidence of full DNA stretching.

b. In addition, it is well known that nanochannel conductance will plateau (due to surface conductance domination) when the salinity of the water decreases below a certain limit, which is widely accepted as the hallmark of nanochannel's unique electrical properties. The authors should demonstrate this to show that the nanochannel fabricated here is of the same properties as others. Another clear tell-tale signature is the generation of ion concentration polarization (ion depletion) in moderate buffer concentrations.

We appreciate the reviewer’s comments. We have modified the manuscript, particularly the abstract and introduction, to clarify our research emphasis and novelty.

a. We have indeed demonstrated single-molecule DNA stretching. The stretching is not achieved within the nanochannels, but in fact by our designed nanopillars arrays. We have modified figure 5 and the manuscript accordingly (pages 12-13).

Figure 5 Single-molecule fluorescence imaging of DNA in sacrificial Si nanochannels. (a) Optical image of nanofluidic regions with nanopillars and nanochannels (40 nm deep, 200 nm wide, 500 nm pitch). (b) Selected fluorescence images showing λ -DNA flowing through nanopillars and nanochannels regions corresponding to the optical images in Figure a. Magenta and yellow dash lines indicate the pillar interface designed for straddling and the nanochannels entry point, respectively. Here frame 1 is defined the first frame the DNA molecule enters the imaged area. The DNA flowed from the bottom to the top. (c) The location-dependent DNA extension due to its hydrodynamic interactions with nanostructures, with the optical graph of the nanofluidic structures added as a location reference. Here the x-axis origin is set as the nanochannel entry. Each black square dot represents the DNA extension in one frame, and the data point of frame 4 is labelled. The time interval between adjacent frames was about 18 msec. The horizontal green dash-dot line indicates the estimated dyed lambda DNA extension when it is fully stretched. Each The scale bar in figure a is 10 μm .

“Consecutive fluorescence images showed that DNA molecules hydrodynamically interacted with diamond-shaped nanopillars through straddling and were consequently stretched (Figure 5), similar to our previous report using a wafer-bonded chip[5]. In our nanostructure design (supplementary Figure S12), the hydrodynamic flow was always 45° to the nanochannels between the diamond-shaped nanopillars, thus forcing DNA to follow a zig-zag path to intentionally promote the straddling interactions. Clearly, the DNA molecule collided onto the nanopillars interface, as indicated by the magenta dash line (Figure 5 a and b), where the designed diamond-shaped pillars transition abruptly from $\sim 3 \mu\text{m}$ wide to about $\sim 1.5 \mu\text{m}$ and the designed pillar gaps decrease from $\sim 700 \text{ nm}$ to $\sim 350 \text{ nm}$. Such a pillar design results in increased DNA fluidic paths and less coiled DNA[7], and as a result a higher probability of DNA straddling onto the pillars.[5] As the hydrodynamic flow moved DNA head and tail forward (Figure 5 b frames 1-4), the DNA molecule became stretched. Then as the head pulled the tail away from the pillar interface (Figure 5 b frames 5-7), a mechanical stress was applied onto the DNA, which kept the DNA in the stretched state. Our measurement results (Figure 5 c) showed that the stretched lambda DNA (48.5 kbp) reached a maximum extension of $\sim 22 \mu\text{m}$ (frame 6), reaching its full contour length with intercalated fluorescence dyes ($\sim 30\%$ longer than without dyes, *i.e.* $\sim 21.5 \mu\text{m}$). The demonstrated DNA stretching through the straddling interactions provides a promising route to linearize not only double-stranded (ds) DNA but also potentially single-stranded (ss) DNA, because this mechanical force based method does not require the design of the nanofluidic structures smaller than the persistence length of DNA ($\sim 50 \text{ nm}$ for ds DNA[8] and $\sim 2 \text{ nm}$ for ss DNA[9]), which are expensive and difficult to manufacture. Previously, we have demonstrated the straddling based DNA stretching by wafer bonding approach, and the successful integration of such functional and complex nanofluidic structures by our sacrificial integration strategy makes it possible to achieve wafer scale manufacturing of single-molecule DNA sensors.[5] In addition, it is also evident that the sacrificial nanofluidic devices are completely compatible with single-molecule fluorescence imaging and very suitable for interrogating the complex biomolecule interactions with nanostructures, making our integration strategy an ideal candidate in a variety of applications where precise, complex, and large-scale integrated nanostructures are necessary for visualizing, manipulating, and detecting biomolecules.[6]”

We also added more information regarding the chip design and DNA translocation in nanochannels in the supplementary documents.

Figure S12 Nanofluidic structure design for DNA straddling demonstration. (a) The overall fluidic structure design of one fluidic branch (700 μm wide), with the nanofluidic structures (magenta) printed by DUV lithography and the connecting microfluidic structures (green) printed by MUV lithography. Note the sacrificial Si fluidic structures are those overlapped by the two layers (blue color), please refer to Figure S5 for integration details (EBL is replaced by DUV for this design). (b-c) Schematics showing detailed nanofluidic structure dimensions.

“Here we aim at demonstrating the capabilities of our sacrificial Si strategy of integrating complex and functional nanofluidic structures. In our design, the nanopillar design is the key to achieving complex DNA hydrodynamic interactions, and the nanofluidic channel dimensions are not critical. The devices were fabricated following the strategy we detailed in previous section (Figure S5), but here we chose DUV lithography rather than EBL to fabricate the nanofluidic structures (Figure S12), similar to our previous report.[5] Each fluidic chip was designed to have six isolated fluidic branches, which have identical pillar designs. Within each fluidic branch (Figure S12 a), the nanofluidic pillars and channels were patterned in an area of 700 μm × 400 μm and connected by microchannels on both sides. The nanofluidic structures included nanochannels in the middle surrounded by symmetrically arranged diamond-shaped nanopillars on each side (Figure S12 b-c). The fluidic design featured diamond-shaped nanopillars with abruptly designed interface to control DNA straddling interaction[10] and pillar gaps that are progressively reduced in dimensions from 1.4 μm to 240 nm, functioning effectively as cascaded two-dimensional fluidic network to pre-stretch the DNA.[7]

“With consecutively captured fluorescence images (exposure time 17.8 ms, frame cycle time 18.1 ms), we studied the single-molecule λ-DNA molecule translocation through the nanopillar and nanochannel

regions (Figure 5). The frame-by-frame speeds and extensions of these DNA molecules were derived by measuring the DNA head and tail locations. The extension L is the fluorescently measured length L_M corrected by DNA travel distance during exposure time τ using measured DNA frame speed v through the relation $L = L_M - v\tau$. The average DNA speed in the imaged region is $140 \mu\text{m}/\text{sec}$. Clearly from Figure 5, the extensions of DNA molecules are strongly correlated to the nanofluidic structure design. In this report, we do not focus on the detailed DNA hydrodynamic interactions with the diamond-shaped pillars, which have thoroughly analyzed in our previous report using similar structures.[5]”

“In a different chip, the DNA molecules have similar hydrodynamic flow, straddling, and relaxation interactions with the nanopillars (Figure S13). Clearly, the DNA molecule stretched much longer after entering the nanochannels (Figure S13 a, frames 6-10) and also straddling nanopillars (Figure S13 a, frames 14-18). This demonstration further illustrates the complexity of DNA hydrodynamic behaviors in nanofluidic structures, and also emphasizes the importance of our integration strategy in nanofluidics and single molecule studies.”

Figure S13 Single-molecule DNA fluorescence imaging and analysis. (a) Representative fluorescence images showing λ -DNA flowing through nanopillars and nanochannels. Yellow dash lines indicate the nanochannels, and magenta dash line indicates the pillar interface for straddling. Here frame 1 is defined the first frame the DNA molecule enters the imaged area. The DNA flowed from the bottom to the top. (b) The location-dependent DNA extension due to its hydrodynamic interactions with nanostructures. Here the origin of the DNA location is set as the nanochannel entry. The time interval between adjacent frames was about 18 msec.

b. The ion concentration dependent conductance in nanopores is an interesting phenomenon and the authors have done similar studies and are fully aware of this.[11] However, the aim of this work is to demonstrate the feasibility of our strategy in wafer-scale integration of complex nanochannels and compatibility with single molecule fluorescence imaging. In this work, the authors are more interested in utilizing the planar nanofluidic channels to interrogate the DNA fluidic dynamic interactions with our designed nanostructures at single-molecule level, and we believe our successful integration strategy could have a big impact in many areas, such as nanofluidics, biomolecule sorting, DNA mapping, sensing and sequencing, etc. In this work, our results of the ionic conductance measurement serve only to

demonstrate the nanochannels wettability and reliability. In the future work, the authors will study in more details of probing biomolecules in nanochannels by looking at the ion conductance change induced by the biomolecules at different buffer concentrations.

4. One of the challenges of the nanochannels, especially the one built by sacrificial layer etching with thin capping layer, is that the filling of very small nanochannel can induce capillary-driven negative pressures within the nanochannel during the filling process, collapsing the nanochannel. Do authors observe this phenomenon? Is this related to the fact that the authors are showing actual DNA data from relatively large (~40nm) channels, not from very small nanochannels of ~10nm or less dimension?

We appreciate the comments regarding the capillary force induced pressure in nanochannels. The following has been added into the manuscript (pages 11-12) to address this concern:

“Structural reliability is an important issue to nanofluidic devices with a thin-film sealing layer, particularly during wetting process. For example, in a slit-like fluidic channel which has a large width/height ratio (as large as 100 [12, 13]), it has been shown the capillary force can induce a large surface tension force to deform and even possibly collapse the sealing layer. Because the vertical mechanical deformation of the capping layer Δh is proportional to its lateral dimension W , a wide and thin capping layer is subject to higher possibility of collapse or breakage, resulting in device failure. This essentially limits the lateral dimension of such nanoslits to smaller than a critical dimension. To mitigate this risk, we utilized a two-dimensional interweaved fluidic network rather than a single wide channel for fluidic transport, and chose nano-scale lateral dimensions for thin (sub 100 nm thick) channels to minimize the width/height ratio. On top of the planar nanochannels, the sealed venting nano-holes also have a small diameter/height ratio (Figure 4a, ~1.5 μm high after sealing, ~300 nm diameter), effectively preventing the sealing SiO_2 layer from collapsing. Therefore, the design of nanometer scale channel and venting hole dimensions is very favorable to mechanical stability of our fluidic chip. Considering the fact that sub 10 nm channels sealed by a thin (1-2 μm) SiO_2 layer could successfully sustain capillary force during DNA flow,[14] we believe our sacrificial nanochannels can also reliably operate at sub 10 nm regime.

“We used capillary force to load DNA into nanochannels (40 nm by 200 nm), similar to our previous work [5]. We did not observe the collapsing of nanochannels or fluidic leakage during channel wetting by either capillary force or by external pressure (>11 Bar), indicating a good mechanical reliability of the sealed channels. The overall mechanical strength of the sacrificial nanofluidic chip is determined by the essentially three-dimensionally structured SiO_2 capping and sealing layers (Supplementary Figure S10), which function as the frame to support the nanofluidic channels and prevent mechanical breakage. To further improve the mechanical robustness, the thickness of the sealing layer could be increased (e.g. from 1 μm in our experiment to 1.5-2 μm), and the area density of venting holes (currently ~300 nm wide, 1 μm and 2 μm pitch in our experiment) next to the critical channel regions can be reduced. Given the flexibility of our integration strategy to accommodate different designs, we believe this sacrificial strategy can be extended to even smaller channels.”

5. While the DNA stretching and manipulation was probably the first application of nanofluidic devices (circa ~2000), it is now widely considered as an alternative membranes (with controlled nanopores). In this regard, it would be important to have not just the accurate nanochannels, but also many of them in parallel, to achieve as a high throughput as typical membranes (nanofiltration membranes, for example). I wonder whether this process can allow stacking more nanofilter layers on top of each other, or other means of generating massively parallel nanochannels.

We appreciate the reviewer's comments.

We agree that nanochannels can function similar to nanopores in single-molecule electronic detection. **The emphasis of the work is to demonstrate the feasibility of our strategy in wafer-scale integration of complex nanochannels and compatibility with single molecule fluorescence imaging on 200 mm wafers.** We agree with the reviewer that high-throughput parallel patterning of multiple nanochannels is very important to many bionanotechnology applications. In fact, the authors did demonstrate multiple nanochannels in parallel as shown in Figure 5 and detailed in supplementary section 5.2. The authors also believe multiple layers of such nanofluidic channels can be stacked together to achieve even more complex three-dimensional fluidic network. This can be achieved by patterning and connecting multiple layers of sacrificial Si structures followed by Si extraction. We have added the text in the manuscript (page 14) to reflect the change.

“It is also envisioned that more complex three-dimensional structures can be integrated using our sacrificial Si strategy, by simply stacking and connecting multiple layers of sacrificial Si structures. This approach could have profound impacts across different fields, such as MEMS, nanophotonics, nanoelectronics, and biosensors.”

6. Lastly, I think authors should discuss one critical issue in this paper, which is the overall cost of the nanochannel fabrication. The lithography and other CMOS compatible processes used in this work requires expensive, unique tools of fabrication, achieving high spatial resolution. It will be expected that the cost of fabrication would be relatively high, although the cost can be reduced in the context of large volume manufacturing, as one can see that such a fabrication can be done economically in volume production. (USB dongles with many gigabytes of flash memory are now much less than 10\$.) I believe that the authors should discuss the overall economics of building the device at different volume scales, since the focus of the paper is on the manufacturing-level fabrication of nanochannel. I think it would be very expensive to make 1 wafers with nanochannels. How many wafers (~1000? ~1,000,000 wafers?) one needs to make to achieve the reasonable cost efficiency of the process?

The authors thank the reviewer for raising this important question. While the work presented in this paper aims to demonstrate a novel progress toward a manufacturing-worthy nanofluidic chip technology, it is too early to give an accurate account of the cost of nanochannel fabrication at this time. We believe the majority of the processes developed at IBM MRL lab are completely compatible with Si

CMOS technology and can be easily transferred to a foundry for large-scale manufacturing. On the other hand, some of the processes in this proof-of-concept demonstration, for example electronic beam lithography (EBL), are used for their versatile design capabilities but not optimized for large-scale production. However, the slow throughput of EBL can be overcome in the future by using alternative nanopatterning methods that are compatible with large-scale electronic device manufacturing, such as DUV lithography, multi-patterning, block-copolymer lithography, etc. With that said, one of the main motivations for us to adopt the CMOS infrastructure to fabricate the nanochannel chips was to capitalize the billions of investment in CMOS chip manufacturing in the IC industry to reduce manufacturing cost of nanofluidic chips, just like the flash memory chips the reviewer correctly pointed out.

In addition, the footprint of the fluidic chips also has a significant impact on the cost per chip. In our current design, the chip has a relatively large size (40 mm) to accommodate a fluorescence microscope available in our DNA characterization lab, which is critical at this stage for imaging single DNA molecules and understanding their fluidic dynamics in our chips. However, the chip size can be greatly reduced once the design has been optimized. Obviously, the fluidic chip dimensions significantly depend on the design and the targeted functionalities, and as a result can vary dramatically from one application to another. Here, our work uses the controlled DNA stretching to demonstrate the compatibility of our integration strategy with various fluidic designs, but it is not practical at this stage to propose a fixed design of the fluidic chips and accordingly calculate the chip cost.

Overall, I believe this is a potentially very impactful paper, but the authors are falling far short of truly convincing the readers that nanochannels can now be made as reliably as typical microelectronics devices. I would hope to see authors to fill this gap and make this work true to its original claims.

The authors thank the reviewer for appreciating the potential of significant impact of our work. This work is a proof-of-concept demonstration of our 200 mm wafer-scale sacrificial Si based integration strategy for functional nanofluidic applications using CMOS compatible fabrication infrastructure. The authors do not claim the proposed sacrificial Si based nanofluidic integration process is as mature as the microelectronics process, which has been developed and refined for almost 70 years dating back to the invention of the first transistor. Rather, the authors demonstrated a viable path of using the existing CMOS based microfabrication technology for fabrication of functional nanofluidic chips, which could be more readily transferred to manufacturing.

The following has been added to the manuscript (page 14):

“Built upon the existing CMOS microfabrication technology, our work demonstrates a 200 mm wafer-scale sacrificial Si based integration strategy for functional nanofluidic applications. Our proof-of-concept demonstration shows a viable path of using the existing CMOS based microfabrication technology for fabrication of functional nanofluidic chips, which could be more readily transferred to manufacturing.”

Reviewer #2 (Remarks to the Author):

Wang et al. in this manuscript present an approach to fabricate nanochannels with critical dimension less than 20 nm at wafer-scale (200 mm) and then investigate its application in single DNA translocation analysis. The fabrication strategy is consistent with standard semiconductor processing, which presents a number of advantages: (1) The monolithic nanofluidic chip integrates microchannels and nanochannels together, which both improves the efficiency of fluidic transport and realizes well-controlled DNA translocation; (2) Robust sealing processes for multilayered nanostructure avoid selective sealing or wafer bonding steps, which means this nanofluidic system can operate at higher pressures and longer times than other designs; (3) Using silicon as a sacrificial material for microchannels and nanochannels avoids wet etching, which often requires longer times and makes it difficult to precisely control the final size and shape of nanochannels; (4) Because it employs a planar (horizontal) fluidic structure rather than a vertical nanopore geometry for studying the single DNA translocation, this design is well-suited to directly observing the motion of DNA molecules in confined nanostructures by fluorescence.

Thus, the approach has a lot going for it, and the paper is well-written.

The authors greatly appreciate the comments by the reviewer. We fully agree with the reviewers on the advantages of our fabrication strategy.

My reservation concerns the fact that the fabrication processes, the layout of the nanofluidic device, and the results of single molecule detection are very similar to previously published work, which is not cited in the manuscript. (IEEE, IEDM13-369-372, 2013 conference). For example, a number of figures in this manuscript, e.g. Fig. S3(c) vs. Fig 3(A); Fig. S8 vs. Fig. 6; Fig. 2(f)-(j) vs. Fig. 8; Fig. 4(a) vs. Fig. 7(B); Fig. 4(d) vs. Fig. 9(A); Fig. 4(e) vs. Fig. 9(c); Fig. 4 (g) vs. Fig. 9(D), and the i-V curves in both papers are almost the same as the previous paper. The numerous similarities raise the issue that the present manuscript is derivative of the earlier conference publication. Considering that Nature Communications requires papers with "important advances of significance to specialists within each field", the authors need to provide a more detailed explanation of the details of this research which speak to its novelty when compared with their previous work. In the absence of such an explanation, I could not recommend publication of the manuscript in Nature Communications in its current form.

The authors thank the reviewer for this important question. The authors believe that we strictly follow the editorial policies of Nature Communications. The authors acknowledge that part of our research progress was presented the 2013 IEEE IEDM conference and disseminated as an IEDM conference abstract. Nonetheless, the authors believe that the publication of part of our work as an IEDM abstract does not compromise novelty per Nature publication policy "Nature journals allow publication of meeting abstracts before the full contribution is submitted." Please refer to Nature duplication policy, <http://www.nature.com/authors/policies/duplicate.html>. Please allow the authors to clarify the difference of the conference publication from this manuscript as follows.

First, the publication format is very different. The conference paper is in fact an **extended abstract** rather than a journal publication. The abstract is limited to 4 pages with constrained space for research details and discussions. Conventionally, the abstracts at conferences are much less cited compared to journal publications in the field of nanofabrication, nanofluidics, and biosensing, because these results

usually provide preliminary information to the audience that was generally limited to progress made at the date of conferences. The authors also noticed the following statement on the Nature communication website (Abstracts or unrefereed web preprints do not compromise novelty).[15]

Second, the review process is very different. The abstract was submitted to the IEDM conference as a standard procedure for all authors who would present their work at the conference. The paper did not have to go through a rigorous peer-review as this manuscript. The authors have not published this work to any peer-reviewed journals.

Third, the data, results, and analyses are different. The authors presented their research progress at IEDM conference for the purpose of rapid communication of preliminary results within part of our research communities. The authors acknowledge that some of the figures look similar, because these are representative results showing our early success during research development. Nonetheless, the authors have added substantially more details, results and analysis regarding design, fabrication, and characterization.

Lastly, we also checked the polices on IEDM. We have attached a pdf file “2013 IEDM Call for Papers”. In this document, **the IEDM conference explicitly states that the extended abstract in no way should preclude the publication in another journal.**

“Authors of accepted papers will be notified by August 13, 2013. They will receive an author's kit, which will include instructions on **the preparation of an extended abstract of no more than 4 pages (including figures) for the Technical Digest of the 2013 IEDM.** This abstract must be submitted to the printer by September 18, 2013. The title of the extended abstract must not be changed from original accepted abstract. **Publication in the digest in no way precludes later publication of a fuller account of the work in another journal,** but NO PUBLICATION is acceptable before the conference.”

Reviewer #3 (Remarks to the Author):

The authors present a wafer-scale fabrication process for producing nanofluidic chips with single-digit nanometer dimensions compatible with standard CMOS processes. The advantages claimed include wafer-scale manufacturing, reliable sealing, and a low temperature process allowing for integration with CMOS circuitry for lab on a chip applications. The presented work is distinguished by the novelty of their process, specifically using sacrificial silicon structures etched by Xenon difluoride gas. The experimental results, observing DNA stretching and translocation, look promising and represent an important potential use of the device.

The advantages claimed seem logical in respect to the current field of research and the work is very well communicated. Overall, the field of micro- and nano-fluidics is very active and this work makes strides in advancing the processes and procedures used.

The authors greatly appreciate the comments by the reviewer. We agree with and thank the reviewers for the comments on the advantages of our work and its significance.

Specific comment:

A large claim of the paper is the wafer-scale capability of the fabrication and its compatibility with traditional CMOS. As mentioned in the paper, yield is a large and important issue in developing CMOS-compatible fluidic structures, yet data (channel conductance measurement) from only one channel from one device is presented. The claims of the paper would be greatly enhanced with data from more than one channel, and preferably, across the devices on the manufactured wafer. If the yield is poor, it would be beneficial for the reader to understand where the process needs further improvement.

We thank the reviewer for the comments. We agree the fabrication uniformity and yield are important issues. We have answered a very similar question in previous section. The answer is also given as follows.

Manufacturing-scale fabrication of these novel devices would significantly facilitate translation of nanofluidic systems into commercialization. This is precisely what we aimed to demonstrate in this work. Our nanofluidic chip design and fabrication processes on 200 mm wafer-scale platform could be directly transferred to a chip foundry for production. We adopted a CMOS compatible fabrication infrastructure, the Microelectronic Research Laboratories (MRL) at the IBM Thomas J. Watson Research Center, to fabricate our nanofluidic chips. The IBM MRL fabrication infrastructure includes Class 100 cleanroom with 200 mm wafer-scale film (metal, semiconductor and dielectric) deposition, 193 nm-, 248 nm- and MUV- optical lithography, e-beam lithography, Reactive Ion Etching, Chemical-Mechanical-Polishing, as well as all relevant metrology and testing tools. The IBM MRL is the birthplace of many groundbreaking innovations in the IC industry, including bipolar technology, SiGe, SOI, Cu interconnects, CMP, and many generations of CMOS technology for several decades. For example, we also demonstrated that IBM MRL is capable of achieving >90% electrical yields consistently from wafer to wafer, see the following figure.[1]

Fig. 4. (Color online) Electrical yields of 45 nm node fatwire level of Cu/248 nm photo-patternable low- κ BEOL structure. These yields were obtained from three line/space test macros, called BEOLCAP macros (BCLP, BLCP_A, and BLCP_B). The nominal feature size of these test macros is 320 nm.

We agree that within-wafer and wafer-to-wafer critical dimension uniformity are key metric for manufacturing-worthy fabrication process. We have added the following to our manuscript (page 6) and supplementary document (section 3.2) to discuss the nanopatterning capability of our integration strategy.

“In this work, we utilize established procedures and recipes at IBM MRL lab during critical nanopatterning steps to maximize the feature uniformity and yield. For example, the plasma etch uniformity in our etch chamber has an etch rate uniformity of within 5% across a 200mm wafer and from wafer-to-wafer. The critical dimension (CD) in DUV lithography has a <15nm variation for a 200nm line/space standard design across a 200 mm wafer, and the yield is about 100% for the dimensions in this work (critical dimension ~200 nm). The high yield is achieved by printing in a controlled and fully automated environment of an ASML and TEL track without manual handling and by applying internal stepper diagnostics on a regular basis to control the focus and dose.”

“In EBL, established CMOS fabrication infrastructure at IBM T.J. Watson research center was used to achieve high resolution and repeatable nanopatterning.[1] To achieve a good control of feature dimensions and maximize the process yields, monitor wafers were added to each batch and processed together with device wafers. This approach enabled us to optimize the recipes for the nanostructure patterning and minimize batch-to-batch and wafer-to-wafer variations. Selected SEM images of the EBL-patterned nanofeatures are shown in supplementary Figure S6.”

Besides manufacturing of fluidic structures at nanometer scale, we would like to emphasize that one key advantage of our sacrificial Si based integration strategy is its capability of integrating complex and functional nanofluidic systems of various dimensions and shapes to fulfill the design requirements of different nanofluidic devices. This has already been discussed in the introduction paragraph of our manuscript, as listed below:

“However, unlike complementary metal-oxide semiconductor (CMOS) chips which utilize a relatively small range of feature dimensions, nanofluidic chips must integrate more sophisticated three-dimensional architectures incorporating vacant and sealed nanostructures with dimensions spanning several orders of magnitude to optimally detect and manipulate biomolecules. The stringent requirement of reliably forming and sealing complex and small nanostructures makes it very challenging to manufacture nanofluidic chips over a wafer scale by current CMOS processes, and thus has seriously hindered electronics integration.”

We also modified our abstract as follows:

“Here we report a scalable 200 mm wafer-scale fabrication strategy capable of producing nanofluidic chips with complex designs and down to single-digit nanometer dimensions. Compatible with industry standard complementary metal-oxide semiconductor (CMOS) logic circuit fabrication processes, this strategy extracts a patterned sacrificial silicon layer through hundreds of millions of nanoscale vent holes on each chip by gas-phase XeF_2 etching, hence dramatically improving the Si extraction yield for large-scale manufacturing. After sealing the vent holes by thin film deposition, the chips can also sustain high pressure (>11 Bar) and long-time operation (>11 h). Verified by single-molecule fluorescence imaging, we demonstrate these sacrificial nanofluidic chips can function to controllably and completely stretch lambda DNA in a two-dimensional nanofluidic network comprising channels and pillars.”

In this work, our results of the ionic conductance measurement serve only to demonstrate the nanochannels wettability and reliability. We have edited the ionic conductance measurement section in supplementary document (section 6) to reflect the changes.

“The aim of this work is to demonstrate the feasibility of our strategy in wafer-scale integration of complex nanochannels and compatibility with single molecule fluorescence imaging. In this work, we carried out ionic conductance measurement using our fluidic probe stations on two randomly selected chips at the edges of the 200 mm wafer.”

“From two wetted fluidic branches on each of the two randomly selected chips, we obtained very similar ionic current (variation <10%). The good agreement is attributed to a few reasons: (1) the nanofluidic structures have uniform dimensions, (2) the nanofluidic channels are fully wet, and (3) the two-dimensional fluidic network in our design has many parallel channels connecting the inlet and outlet, and hence has a much more stable current compared to a single channel. The small variation is attributed to occasional air bubbles injected by the fluidic probes, interface resistance at the Ag/AgCl electrodes, etc.”

In the future work, the authors will study in more details of probing biomolecules in nanochannels by looking at the ion conductance change induced by the biomolecules at different buffer concentrations.

References:

- [1] Q. Lin, A. Nelson, S.-T. Chen, P. Brock, S. A. Cohen, B. Davis, *et al.*, "Integration of Photo-Patternable Low- κ Material into Advanced Cu Back-End-Of-The-Line," *Japanese Journal of Applied Physics*, vol. 49, p. 05FB02, 2010.
- [2] H. Winters and J. Coburn, "The etching of silicon with XeF₂ vapor," *Applied Physics Letters*, vol. 34, pp. 70-73, 1979.
- [3] M. Vugts, M. Eurlings, L. Hermans, and H. Beijerinck, "Si/XeF₂ etching: Reaction layer dynamics and surface roughening," *Journal of Vacuum Science & Technology A*, vol. 14, pp. 2780-2789, 1996.
- [4] X. T. Huang, C. Gupta, and S. Pennathur, "A novel fabrication method for centimeter-long surface-micromachined nanochannels," *Journal of Micromechanics and Microengineering*, vol. 20, p. 015040, 2009.
- [5] C. Wang, R. L. Bruce, E. A. Duch, J. V. Patel, J. T. Smith, Y. Astier, *et al.*, "Hydrodynamics of diamond-shaped gradient nanopillar arrays for effective DNA translocation into nanochannels," *Acs Nano*, vol. 9, pp. 1206-1218, 2015.
- [6] B. H. Wunsch, J. T. Smith, S. M. Gifford, C. Wang, M. Brink, R. Bruce, *et al.*, "Nanoscale Lateral Displacement Arrays for Separation of Exosomes and Colloids Down to 20nm," *Nature nanotechnology*, p. DOI: 10.1038/NNANO.2016.134, 2016.
- [7] W. Reisner, K. J. Morton, R. Riehn, Y. M. Wang, Z. N. Yu, M. Rosen, *et al.*, "Statics and dynamics of single DNA molecules confined in nanochannels," *Physical Review Letters*, vol. 94, p. 196101, May 2005.
- [8] W. Reisner, J. N. Pedersen, and R. H. Austin, "DNA confinement in nanochannels: physics and biological applications," *Reports on Progress in Physics*, vol. 75, p. 106601, 2012.
- [9] B. Tinland, A. Pluen, J. Sturm, and G. Weill, "Persistence Length of Single-Stranded DNA," *Macromolecules*, vol. 30, pp. 5763-5765, 1997/09/01 1997.
- [10] Y. Viero, Q. H. He, and A. Bancaud, "Hydrodynamic Manipulation of DNA in Nanopost Arrays: Unhooking Dynamics and Size Separation," *Small*, vol. 7, pp. 3508-3518, Dec 2011.
- [11] S.-W. Nam, M. J. Rooks, K.-B. Kim, and S. M. Rossnagel, "Ionic field effect transistors with sub-10 nm multiple nanopores," *Nano letters*, vol. 9, pp. 2044-2048, 2009.
- [12] N. R. Tas, P. Mela, T. Kramer, J. Berenschot, and A. van den Berg, "Capillarity induced negative pressure of water plugs in nanochannels," *Nano Letters*, vol. 3, pp. 1537-1540, 2003.
- [13] N. R. Tas, M. Escalante, J. W. van Honschoten, H. V. Jansen, and M. Elwenspoek, "Capillary negative pressure measured by nanochannel collapse," *Langmuir*, vol. 26, pp. 1473-1476, 2010.
- [14] Q. Xia, K. J. Morton, R. H. Austin, and S. Y. Chou, "Sub-10 nm self-enclosed self-limited nanofluidic channel arrays," *Nano letters*, vol. 8, pp. 3830-3833, 2008.
- [15] N. Communications. (07/21). *HOW TO SUBMIT*. Available: <http://www.nature.com/ncomms/authors/submit.html>

Reviewer #1 (Remarks to the Author):

The authors have responded to the reviewer's criticisms in the revised paper. But I still have one major issue, plus several minor points about this paper, which prevents me from supporting the publication of this paper.

Major Issue: The authors' response to my comment (as well as the reviewer #3) on the issue of yield, uniformity, and variation is unacceptable. Simply put, we do not have any information from this paper regarding the yield, uniformity, reliability of THIS SPECIFIC PROCESS, and the authors are providing generic arguments about the reliability of established CMOS process in general. From this paper, I wonder if;

1. The authors have so far created just one successfully fabricated chip without any defect, out of many 200mm wafers attempted.
2. The authors have so far created one successful wafer, with good uniformity across the entire wafer that was successful, but with significant wafer-to-wafer variations.
3. The authors have established many successfully fabricated wafers with the same uniformity and quality across the board.

What is the case in this work? The authors simply claims the great uniformity of the CMOS process, which I agree. But they do not provide any information regarding the yield of this specific devices and processes. In typical reviews, we tend not to focus too much on this issue of yield, but in this paper, this information is critical.

We great appreciate the reviewer's comments and admire the reviewer's persistence. It is such critical thinking from reviewers such as this that help *Nature Communications* maintain a high publication standard. As discussed previously, reliable manufacturing of nanofluidic devices have never been demonstrated before. We aim to demonstrate 200 mm scale-scale fabrication of these novel devices to facilitate translation of such nanofluidic systems into commercialization. Yield, uniformity, and variation are key parameters for a manufacturing worthy fabrication processes.

We would like to clarify that the work presented in this manuscript is research and development in nature, not high-volume manufacturing or production. To avoid confusion, we have replaced "manufacture," "manufacturing", "production" with "fabricate" or "fabrication" in the manuscript.

In our previous response, we showed that the facility/infrastructure we used to fabricate our 200 mm wafer-scale nanofluidic chips is capable of making thousands of IC-type chips on multiple wafers with >90% **electrical** yields. We have used the same facility/infrastructure to make multiple wafers (each with 12 nanofluidic chips) and tested multiple chips for fluidic functions one chip at a time. We examined extensively the critical dimension (CD) of the patterns fabricated and found that we were able to consistently and reliably fabricate nanochannels with CD down to 20 nm. Unfortunately we do not have the wafer-scale fluidic testing capability to generate the same **fluidic** yield data as the electrical yield data with the IC-type chips.

We have added the following to the manuscript (page 11).

“We observed that our integration and fabrication strategy allowed us to consistently and reliably fabricate functional nanochannels with critical dimensions down to 20 nm on multiple 200 mm wafers with 12 nanofluidic chips per wafers.”

This is because, as the authors are pointing out as well, all of the scientific concepts have been already known and published before, including nanochannels and their DNA stretching based sensing, nanochannel fabrication by XeF₂ sacrificial etching, and even the use of access holes (followed by cover-up/resealing) to facilitate the removal of sacrificial layer (see S. W. Turner et al., J. Vac. Sci. Technol., B 16, 3835 (1998)). The only major/novel claim is the wafer-scale integration of these tricky processes, but I don't see any evidence of such "wafer-scale integration" in this work. If the yield is limited, then (as the third reviewer points out) the authors at least should try to point out the reasons, to really advance the field as typical Nature-level papers do.

The author appreciate the comments. We agree our research success builds on previous scientific knowledge, as any ground-breaking research does. However, we disagree with the reviewer that our work lacks of novelty. **In fact, we have already addressed the difference of our approach from others' (page 4).** Briefly, our approach significantly differs from previous methods in its critical structure dimension, fluidic structure design complexity, compatibility with single-molecule imaging, ease of fabrication, and capability of wafer-scale integration, etc. This paragraph is highlighted one more time in the text.

“XeF₂ etching of Si is a well-established technique with demonstrated compatibility with Si processing.^{1,2} However, to date, its applications in complexed designed functional nanofluidics have been rather limited.³ Our demonstration significantly differs from others XeF₂ based integration methods. First, previous demonstrations usually have large lateral dimensions, e.g. from 10 to 100 μm³. In contrast, the critical dimensions of our devices are about 3 orders of magnitude smaller. Second, single-molecule imaging and manipulation (e.g. DNA stretching), which is important to biomolecular sensing, sorting, etc., has not been demonstrated previously in sacrificial nanochannels. Here we visualize fluorescently labelled single DNA molecule flow and verify their controlled stretching in nanochannels. Third, previous XeF₂ Si etching was only applied to simple geometries such as straight and long channels but not complex and functional fluidic network. In this work, we prototype two-dimensional fluidic network for controlled DNA fluidic dynamics and stretching. Fourth, conventional methods diffuse XeF₂ only through the fluidic ports to extract Si, which is inherently a time-consuming diffusion-limited process and strongly dependent on the channel dimensions. Differently, we rationally integrate venting holes to initiate the Si exaction in a parallel fashion, hence significantly reducing the process time, increasing the throughput, and enabling complex fluidic design over large areas. Lastly, we integrate our process completely on a 200 mm Si wafer processing platform, making it easier for our integration strategy to translate to high-volume production.”

In fact, we already made a comparison of different sacrificial methods, and we already cited and referred to publication mentioned by the reviewer (page 3), see below. Noticeably, **our method uses vapor phase**

etching of sacrificial Si material, and the referenced paper used wet etching. The two methods are fundamentally different.

“Sacrificial approaches utilize a material “to be sacrificed” patterned into a reverse image of the desired nanofluidic structures, and selectively extract this sacrificial material at a later stage of processing to form the nanofluidic system. However, thermal decomposition based extraction method⁴ has serious risks of structural damage at elevated temperatures, and wet etching based extraction processes⁵⁻⁷ are ineffective at nanometer scales and potentially destructive, because removing etched byproduct becomes exceedingly difficult and undesirable long processing time is needed⁷.”

Minor issue

In response to my comments, the authors greatly expanded the discussion on the design of venting hole, and how one can even fabricate very small nanochannels by properly locating the holes on a nonnanochannel section. They also argue that their focus is on short, one dimensional nanochannel fabrication for the specific goal of DNA stretching. I would like to remind the authors again that the main claim of this work is the process, not the particular chip (if it were only about linear nanochannel fabrication, the impact of the work would be diminished). I would encourage authors to add more design limitation discussions, to help the readers when this process can or cannot be used, and how. Specifically, what is the practical limitation of nanochannel width vs. distance of nanochannel from access hole vs. etching time allowed (given the etch selectivity)? Such discussions are essential for a paper claiming to report novel/innovative processes. It is probably OK for authors not being able to make extremely long and narrow nanochannels, but I still think it is authors' obligation to clearly tell what can and cannot be made using their process.

The authors thank the reviewer's comments. However, the reviewer missed the key points emphasized in this paper. **This paper does NOT focus on “short, one dimensional nanochannel fabrication”, in fact, we clearly demonstrate the integration of complexly designed two-dimensional nanofluidic network to achieve our desired nanofluidic functionality.** In this work, we use the controlled DNA stretching in such two-dimensional nanofluidic network as a demonstration, but the application of our method is NOT only limited to DNA stretching.

The authors also want to clarify again that extremely long and narrow channels are not designed in this work because they are irrelevant to our desired DNA stretching demonstration. In addition, we believe narrow and long channels can be designed using our sacrificial channel method. For example, we have shown 13 nm wide and 20 μm long channels can be extracted (manuscript page 9 and supplementary Figure S8). Further, we also emphasize that making long nanochannels does not always benefit the applications, because the wetting process rather than fabrication can be a very serious challenge. Due to the capillary force, once air bubbles are trapped in such channels, they will be very difficult to remove. Therefore, we would rather design more complexly designed two-dimensional nanofluidic network in this work, and we don't have an intention to go to the extreme width and length in our design.

“One key focus of this work is to demonstrate the compatibility of **our integration strategy with complex functional two-dimensional fluidic networks** (supplementary Figure S8). Accordingly, the mesh-like channels (~400 μm long) are designed much longer than the straight channel portion (5 to 20 μm) to focus on the DNA interactions with the nanopillars. **Notably, our integration strategy has no limitation to the lengths of such fluidic network and can be generalized to a variety of nanofluidic designs.** Despite the fact that the etching rate drops with narrow channels, long sacrificial Si channels can be extracted by either increasing the XeF_2 etching cycles or by adding nanometer-scale venting holes on top of the channel when they are desired in design. **On the other hand, narrow, long, and isolated nanochannels have constrained applications because their wetting can be even more challenging than the fabrication. In this work, unnecessarily narrow and long nanochannels were not needed to achieve our desired DNA stretching in our two-dimensional fluidic network.**”

Regarding the question “the practical limitation of nanochannel width vs. distance of nanochannel from access hole vs. etching time allowed”, we have already included a figure (supplementary Figure S8) to demonstrate the extraction of narrow channels as small as 13 nm and explicitly shown the etching time dependence on the nanochannel width. We also modified the following discussion to address this issue (page 9):

“For example, to minimize possible damage to the critical nanostructures during venting hole formation and sealing process, we intentionally avoided nanoholes in a region spanning up to 60 μm long. Such a hole-free region was designed to consist of 20 μm long narrow channels and ~20 μm long two-dimensional fluidic network on each side of the channels, and can be flexibly designed and easily adjusted in DUV lithography. The large optical reflection difference between Si and SiO_2 allows reliable process monitoring during XeF_2 etching processes (Figure 3 d-f) to ensure complete removal of sacrificial Si. Importantly, we also noticed that XeF_2 can diffuse and remove sacrificial Si nanostructures in as small as 13 nm nanochannels that are not directly connected to venting holes (supplementary Figure S8). However, the XeF_2 Si etching is slower at smaller nanostructured Si dimensions, because the vapor-phase transport of the XeF_2 precursor to Si surface and the volatile byproducts away from Si are strongly dependent on channel dimensions. Since our designed critical nanochannels 13 to 67 nm, supplementary Figure S8) are much narrower than the fluidic network connecting to them (>240 nm, details in supplementary Figure S12), XeF_2 is expected to diffuse much slower in the narrow nanochannels. Accordingly the etching time required to completely extract Si is not determined by the distance of nanochannels from the access holes but rather by the widths and lengths of narrow nanochannels. In principle, any nanochannels can be eventually extracted given long enough etching time, however, in practice specific design and manufacturing requirements may limit how long the extraction process can be and how narrow the nanochannel can be designed. “

Reviewer #2 (Remarks to the Author):

The authors have submitted a revised manuscript which addresses a number of issues. However, they have not cited the previous conference proceeding identified in the original review this despite their own persuasive arguments that the earlier proceedings paper does not compromise the novelty or need/ability to publish in a full paper. Although this is a strong paper in the main, in this point, at least, the revision is not responsive to the review.

The authors greatly appreciate the reviewer's time and efforts during the reviewing process, and have referenced the conference proceeding in the manuscript (page 4).

“Our integration strategy was first disseminated in a conference abstract,⁸ but this paper provides for the first time detailed and complete discussions on the integration strategy, key challenging issues, fabrication results, and single molecule DNA straddling.”

Reviewer #3 (Remarks to the Author):

The authors have comprehensively and constructively addressed all of my previous comments. I would like to recommend publication of this paper in Nature Communications.

The authors greatly appreciate the reviewer's time and efforts during the reviewing process.

Reference:

- 1 Winters, H. & Coburn, J. The etching of silicon with XeF₂ vapor. *Appl. Phys. Lett.* **34**, 70-73 (1979).
- 2 Vugts, M., Eurlings, M., Hermans, L. & Beijerinck, H. Si/XeF₂ etching: Reaction layer dynamics and surface roughening. *J. Vac. Sci. Technol. A* **14**, 2780-2789 (1996).
- 3 Huang, X. T., Gupta, C. & Pennathur, S. A novel fabrication method for centimeter-long surface-micromachined nanochannels. *Journal of Micromechanics and Microengineering* **20**, 015040 (2009).
- 4 Li, W. *et al.* Sacrificial polymers for nanofluidic channels in biological applications. *Nanotechnology* **14**, 578 (2003).
- 5 Turner, S., Perez, A., Lopez, A. & Craighead, H. Monolithic nanofluid sieving structures for DNA manipulation. *J. Vac. Sci. Technol., B* **16**, 3835-3840 (1998).
- 6 Lee, C., Yang, E.-H., Myung, N. V. & George, T. A nanochannel fabrication technique without nanolithography. *Nano Lett.* **3**, 1339-1340 (2003).
- 7 Stern, M. B., Geis, M. W. & Curtin, J. E. Nanochannel fabrication for chemical sensors. *J. Vac. Sci. Technol., B* **15**, 2887-2891 (1997).
- 8 Wang, C. *et al.* in *IEEE International Electron Devices Meeting (IEDM)* DOI: 10.1109/IEDM.2013.6724627.

We appreciate the time by referee #1 to review the revised manuscript. We believe that we have clearly emphasized the novelty and the impact of our technology in previous revision reports, which have been well received by the other two referees. We recognize that referee #1 has a strong opinion on our work which we respect. We would not seek to convince referee #1 but would leave the general readers of the Nature Communications to evaluate the importance of our work. We thank the critical comments from referee #1 very much, which helped us tremendously to improve the quality of this paper.

Reviewer #1 (Remarks to the Author):

I have reviewed the second revision of Wang et al. and, after a long time debating with myself, my conclusion is that this paper, at least in my opinion, is not representing the advances and breakthroughs we typically associate with journals like Nature Communication. On one hand, I do think strongly that more emphasis and credit should be given to the technological breakthrough (not just scientific outcomes), since those who are working on the technology side of things tend not to get the justified credit. But, on the other hand, in my judgement, many authors of the previous papers cited in this paper (related to the earlier implementation of the fabrication technology described in this paper) will find this work to be rather incremental and not really advancing things far ahead from what they have achieved in the past.

1. Based on the responses, it appears that the authors did not reach the level of "reliability testing" that are often associated with the industrially implemented process. They are offering to change the title from "manufacturing" to "fabrication". But, "fabrication" of nanochannels described in this paper has been achieved using other techniques. There is no doubt that new ideas are used in this work, but I don't think the level of innovation described in this paper can be called "transformative" or "really advancing the field".

2. The fabrication of nanochannel demonstrated in this work is impressive, but not drastically different from earlier implementation for DNA stretching / mapping applications. For those applications, you only need ~50micron or less long 1D nanochannels, and the authors are demonstrating the techniques (how to design access holes, etc.) for both 1D nano channel and the loading and unloading channels (this can

be considered as 2D nanochannel). In my judgement, this was an (important but not necessarily ground-breaking) extension of previous sacrificial etching based nanofilter fabrications papers.